# COMPOSITIONAL VIDEO GENERATION AS FLOW EQUALIZATION

## ABSTRACT

Large-scale Text-to-Video (T2V) diffusion models have recently demonstrated unprecedented capability to transform natural language descriptions into stunning and photorealistic videos. Despite these promising results, a significant challenge remains: these models struggle to fully grasp complex compositional interactions between multiple concepts and actions. This issue arises when some words dominantly influence the final video, overshadowing other concepts. To tackle this problem, we introduce **Vico**, a generic framework for compositional video generation that explicitly ensures all concepts are represented properly. At its core, Vico analyzes how input tokens influence the generated video, and adjusts the model to prevent any single concept from dominating. Specifically, Vico extracts attention weights from all layers to build a spatial-temporal attention graph, and then estimates the influence as the *max-flow* from the source text token to the video target token. Although the direct computation of attention flow in diffusion models is typically infeasible, we devise an efficient approximation based on subgraph flows and employ a fast and vectorized implementation, which in turn makes the flow computation manageable and differentiable. By updating the noisy latent to balance these flows, Vico captures complex interactions and consequently produces videos that closely adhere to textual descriptions. We apply our method to multiple diffusion-based video models for compositional T2V and video editing. Empirical results demonstrate that our framework significantly enhances the compositional richness and accuracy of the generated videos.

## 1 INTRODUCTION

Humans recognize the world compositionally. That is to say, we perceive and understand the world by identifying parts of objects and assembling them into a whole. This ability to recognize and recombine elements—making "infinite use of finite mean"—is crucial for understanding and modeling our environment. Similarly, in the realm of generative AI, particularly in video generation, it is crucial to replicate this compositional approach.

Despite advancements in generative models, current diffusion models fail to capture the true compositional nature of inputs. Typically, some words disproportionately influence the generative process, leading to visual content that does not reflect the intended composition of elements. While the compositional text-to-image sythesis (Liu et al., 2022; Chefer et al., 2023; Kumari et al., 2023; Feng et al., 2023; Huang et al., 2023) has been more studied, the challenge of compositional video generation has received less attention. This oversight is largely due to the high-dimensional nature of video and the complex interplay between concepts and motion.

As an illustration, we highlight some failure cases in Figure 1 (Left), where *certain words dominate* while others are underrepresented. Common issues include *missing subject* and *spatial confusion*, where some concepts do not appear in the video. Even with all concepts present, *semantic leakage* can occur, causing attributes amplified incorrectly, for example, the prompt of *a bird and a cat* is misinterpreted as *a bird looks like a cat*. A challenge specific to T2V is *Motion Mixing*, where the action intended for one object mistakenly interacts with another, such as generating a `flying wale` instead of `flying balloon`.

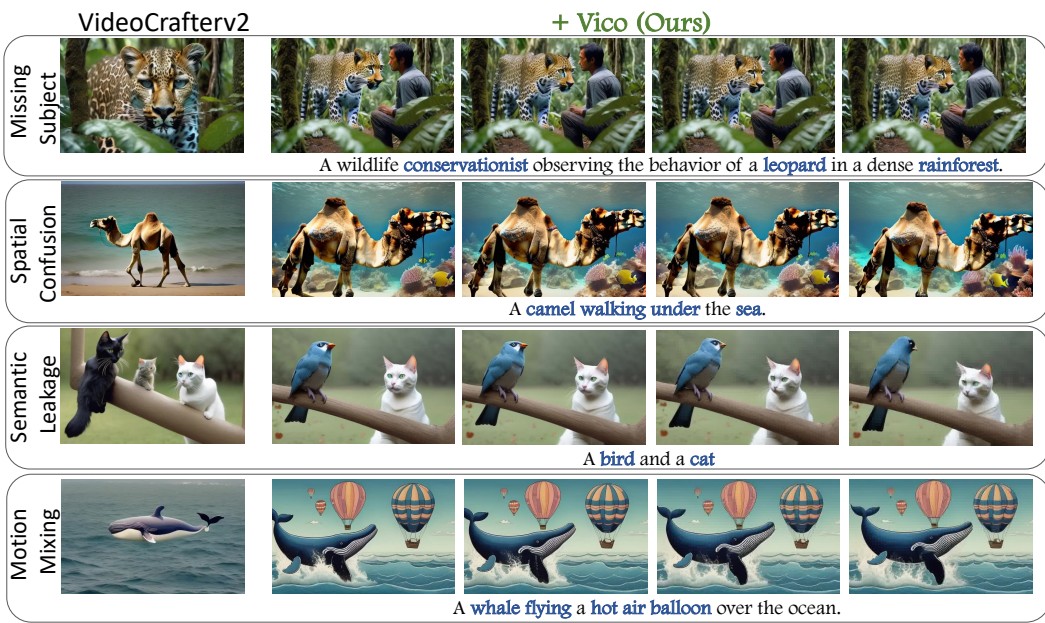

Figure 1: Examples for compositional video generation of **Vico** on top of VideCrafterv2 (Chen et al., 2024). We identify four types of typical failure in compositional T2V (Row 1) *Missing Subject* (Row 2) *Spatial Confusion* (Row 3) *Semantic Leakage* and (Row 4) *Motion Mixing*. **Vico** provides a unified solution to these issues by equalizing the contributions of all text tokens.

To address these challenges, we present **Vico**, a novel framework for compositional video generation that ensures all concepts are represented equally. Vico operates on the principle that, each textual token should have an equal opportunity to influence the final video output. At our core, Vico first assesses and then rebalances the influence of these tokens. This is achieved through test-time optimization, where we assess and adjust the impact of each token at every reverse time step of our video diffusion model. As shown in 1, Vico resolves the above questions and provides better results.

One significant challenge is accurately attributing text influence. While cross-attention (Tang et al., 2023; Mokady et al., 2022; Feng et al., 2023; Rassin et al., 2024) provides faithful attribution in text-to-image diffusion models, it is not well-suited for video models. It is because such cross-attention is only applied on spatial modules along, treating each frame independently, without directly influencing temporal dynamics.

To surmount this, we develop a new attribution method for T2V model, termed *Spatial-Temporal Attention Flow* (**ST-flow**). ST-flow considers all attention layers of the diffusion model, and views it as a spatiotemporal flow graph. Using the maximum flow algorithm, it computes the flow values, from input tokens (sources) to video tokens (target). These values serve as our estimated contributions.

Unfortunately, this naive attention max-flow computation is, in fact, both computationally expensive and non-differentiable. We thus derive an efficient and differentiable approximation for the ST-Flow. Rather than computing flow values on the full graph, we instead compute the flow on all subgraphs. The ST-Flow is then estimated as the maximum subgraph flow. Additionally, we have develop a special matrix operation to compute this subgraph flow in a fully vectorized manner, making it approximately $100\times$ faster than the exact ST-flow.

Once we obtain these attribution scores, we proceed to optimize the model to balance such contributions. We do this as a min-max optimization, where we update the latent code, in the direct that, the least represented token should increase its influence.

We implement Vico on multiple video applications, including text-to-video generation and video editing. These applications highlight the framework's flexibility and effectiveness in managing complex prompt compositions, demonstrating significant improvements over traditional methods in both the accuracy of generated video. Our contributions can be summarized below:

- We introduce **Vico**, a framework for compositional video generation. It optimizes the model to ensure each input token fairly influences the final video output.

- We develop ST-flow, a new attribution method that uses attention max-flow to evaluate the influence of each input token in video diffusion models.

- We derive a differentiable method to approximate ST-flow by calculating flows within subgraphs. It greatly speed up computations with a fully vectorized implementation.

- Extensive evaluation of Vico in diverse settings has proven its robust capability, with substantial improvements in video quality and semantic accuracy.

## 2 PRELIMINARIES

**Denoising Diffusion Probabilistic Models.** Diffusion model reverses a progressive noise process based on latent variables. Given data $\mathbf{x}_0 \sim q(\mathbf{x}_0)$ sampled from the real distribution, we consider perturbing data with Gaussian noise of zero mean and $\beta_t$ variance for $T$ steps/ At the end of day, $\mathbf{x}_T \rightarrow \mathcal{N}(0, \mathbf{I})$ converge to isometric Gaussian noise. The choice of Gaussian provides a close-form solution to generate arbitrary time-step $\mathbf{x}_t$ through

$$\mathbf{x}_t = \sqrt{\bar{\alpha}_t}\mathbf{x}_0 + \sqrt{1 - \bar{\alpha}_t}\boldsymbol{\epsilon}, \quad \text{where} \quad \boldsymbol{\epsilon} \sim \mathcal{N}(0, \mathbf{I}) \tag{1}$$

where $\alpha_t = 1 - \beta_t$ and $\bar{\alpha}_t = \prod_{s=1}^{t} \alpha_s$. A variational Markov chain in the reverse process is parameterized as a time-conditioned denoising neural network $\boldsymbol{\epsilon}_{\boldsymbol{\theta}}(\mathbf{x}, t)$ with $p_{\boldsymbol{\theta}}(\mathbf{x}_{t-1}|\mathbf{x}_t) = \mathcal{N}(\mathbf{x}_{t-1}; \frac{1}{\sqrt{1-\beta_t}}(\mathbf{x}_t + \beta_t\boldsymbol{\epsilon}_{\boldsymbol{\theta}}(\mathbf{x}, t)), \beta_t\mathbf{I})$. The denoiser is trained to minimize a re-weighted evidence lower bound (ELBO) that fits the noise

$$\mathcal{L}_{\text{DDPM}} = \mathbb{E}_{t, \mathbf{x}_0, \boldsymbol{\epsilon}}\left[||\boldsymbol{\epsilon} + \sqrt{1 - \bar{\alpha}_t}\boldsymbol{\epsilon}_{\boldsymbol{\theta}}(\mathbf{x}, t)||_2^2\right] \tag{2}$$

Training with denoising loss, $\boldsymbol{\epsilon}_{\boldsymbol{\theta}}$ equivalently learns to recover the derivative that maximize the data log-likelihood (Song & Ermon, 2019; Hyvärinen & Dayan, 2005; Vincent, 2011). With a trained $\boldsymbol{\epsilon}_{\boldsymbol{\theta}^*}(\mathbf{x}, t) \approx \nabla_{\mathbf{x}_t} \log p(\mathbf{x}_t)$, we generate the data by reversing the Markov chain

$$\mathbf{x}_{t-1} \leftarrow \frac{1}{\sqrt{1 - \beta_t}}(\mathbf{x}_t + \beta_t\boldsymbol{\epsilon}_{\boldsymbol{\theta}^*}(\mathbf{x}, t)) + \sqrt{\beta_t}\boldsymbol{\epsilon}_t; \tag{3}$$

The reverse process could be understood as going along $\nabla_{\mathbf{x}_t} \log p(\mathbf{x}_t)$ to maximize the likelihood.

**Text-to-Video (T2V) Diffusion Models.** Given a text prompt $y$, T2V diffusion models progressively generate a video from Gaussian noise. This generation typically occurs within the latent space of an autoencoder (Rombach et al., 2022) to reduce the complexity. The architecture design of T2V models often follows either a 3D-UNet (Ho et al., 2022b; Blattmann et al., 2023b; Ho et al., 2022a; Harvey et al., 2022; Wu et al., 2023a) or diffusion transformer (Gupta et al., 2023; Peebles & Xie, 2023; Ma et al., 2024). For computational efficiency, these architectures commonly utilize separate self-attention (Vaswani et al., 2017) for spatial and temporal tokens. Moreover, cross-attentions is applied on each frame separately, thereby injecting conditions into the model. More related work is in Appendix C.

**Maximum-Flow Problem.** (Harris & Ross, 1955; Ford & Fulkerson, 1956; Edmonds & Karp, 1972) Consider a directed graph $G(V, E)$ with a source node $s$ and a target node $t$. A flow is function on edge $f : E \rightarrow \mathbb{R}$ that satisfies both *conservation constraint* and *capacity constraint* at every vertex $v \in V \setminus \{s, t\}$. This means the total inflow into any node $v$ must equals its total outflow, and the flow on any edge cannot exceed its capacity. The flow value $|f| = \sum_{e_{s,v} \in E} f(s, u)$ is defined as the total flow out of the source $s$, which is equal to the total inflow into the target $t$, $|f| = \sum_{e_{u,t} \in E} f(u, t)$. The maximum flow problem is to find a flow $f^*$ that maximizes this value.

## 3 VICO: COMPOSITIONAL VIDEO GENERATION AS FLOW EQUALIZATION

In this paper, we solve the problem of compositional video generation by equalizing influence among tokens. We calculate this influence using max-flow within the attention graph of the T2V model and ensure efficient computation. We define our problem and optimization scheme in Sec 3.1. The definition of ST-Flow and its efficient computation are discussed in Sections 3.2 and 3.3.

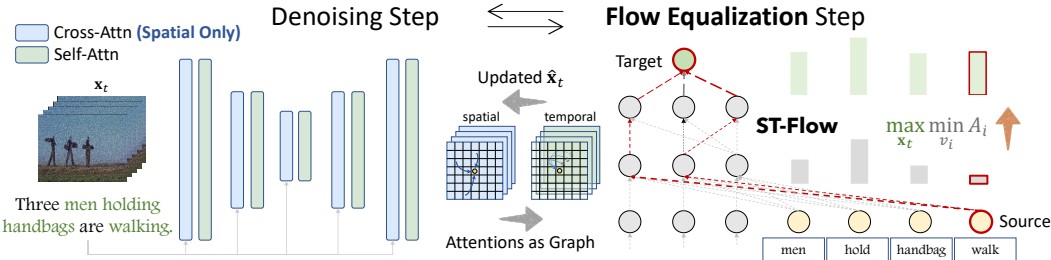

Figure 2: Overall pipeline of our **Vico**. Before each denoising step, Vico extracts attention maps from each layer to build a spatiotemporal graph. We calculate the attribution scores as max-flow in the graph and adjust the noisy latent code to balance this flows.

### 3.1 OVERALL PIPELINE AND OPTIMIZATION

Our goal is to generate a video from a given input prompt $P$. Rather than focusing on all tokens in the prompt, we target a subset of $K$ key tokens of interest, $\mathcal{V} = v_1, \ldots, v_K$, such as subjects and verbs. We aim to ensure that those token fairly contributes to the final video. This process is detailed in Figure 2.

**Objectives.** To achieve this, we define an attribution function $A_i = A(v_i) \in \mathbb{R}$ for each token $v_i$. Intuitively, $A_i$ represents the importance for each token within the model, quantifying its impact on the video. We optimize the attribution scores to ensure fairness:

$$\max_{\mathbf{x}_t} \mathcal{L}_{\text{fair}}(A_1, \ldots, A_K) = \max_{\mathbf{x}_t} \min_{v_i} \{A_1, \ldots, A_K\}; \qquad (4)$$

Here, $\mathcal{L}_{\text{fair}} = \min_{v_i}\{A_1, \ldots, A_K\}$ serves as the fairness function, focusing on the least represented token. By updating the noisy latent $\mathbf{x}_t$ to maximize $\mathcal{L}_{\text{fair}}$, we ensure equal contributions across all tokens. The measurement of $A_i$ could be general. Specific to our paper, we estimate $A_i$ as flow in attention graph, which will be discussed in Section 3.2.

**Optimization.** To implement Eq 4, we perform test-time optimization. Before each denoising step, we first feed $\mathbf{x}_t$ into the model, extract the $A_i$, and update $\mathbf{x}_t$ through gradient ascent: $\hat{\mathbf{x}}_t \leftarrow \mathbf{x}_t + \eta \nabla_{\mathbf{x}_t} \mathcal{L}_{\text{fair}}(A_1, \ldots, A_K)$. $\eta$ is the step size. Then, $\hat{\mathbf{x}}_t$ is going through a denoising step to get $\mathbf{x}_{t-1}$ according to Eq 3. We repeat these steps until the video is generated.

### 3.2 ATTENTION FLOW ACROSS SPACE AND TIME

With above formulation, our focus is to develop an efficient and precise attribution $A_i$. Recognizing issues with cross-attention, we instead calculate $A_i$ as the flow through the entire attention graph.

**Flawed Cross-Attention in Text-to-Video Models.** Cross-attention score has been instrumental in attributing (Tang et al., 2023) and controlling layout and concept composition in text-to-image models (Hertz et al., 2022; Chefer et al., 2023; Rassin et al., 2024). However, applying it to T2V diffusion model introduces new problem.

This problem arises because T2V models typically employ cross-attention on spatial tokens only (Wang et al., 2023a; Chen et al., 2023; Wang et al., 2023b). It treats the video as a sequence of independent images, and temporal self-attention mixes tokens across different frames. Consequently, this separation hinders cross-attention's ability to capture video dynamics, making it challenging to manage actions across frames.

For example, applying the cross-attention-based DAAM attribution (Tang et al., 2023) on VideoCrafterv2 reveals significant issues in visualization. As shown in Figure 3 (Left), cross-attention leads to a flickering pattern in the attention maps, failing to consistently highlight the target object across frames.

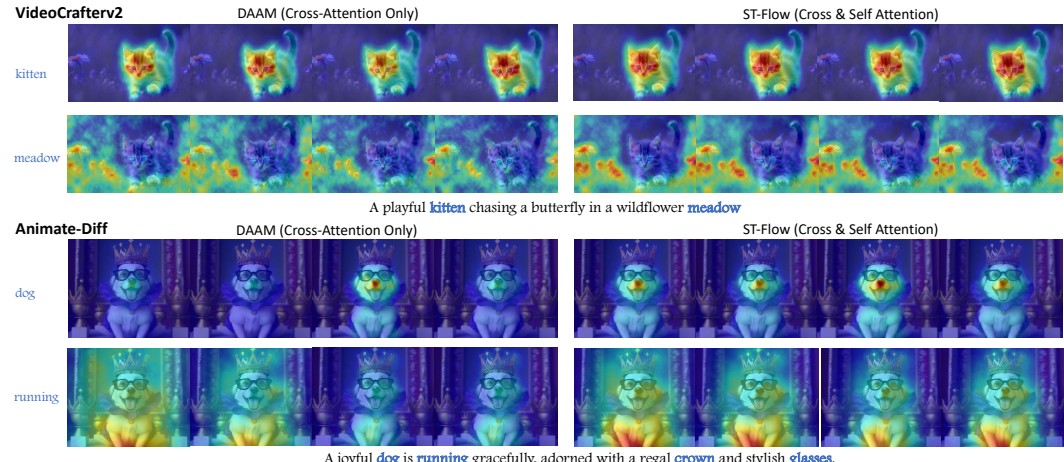

Figure 3: Attribution heatmap comparison between DAAM and our ST-Flow.

Recognizing these limitations, we propose a new measurement termed *Spatial-Temporal Flow (ST-Flow)*, which estimates the influence throughout the entire spatiotemporal attention graph in the video diffusion model. As seen in Figure 3 (Right), ST-Flow gets heatmap with improved consistency.

**Attention as a Graph Over Space and Time.** In our approach, we conceptualize the stacked attention layers as a directed graph $G = (V, E)$, where nodes represent tokens and edges weighted by the influence between tokens. A 4-layer example is illustrated in Figure 2 (Right).

Its adjacency matrix is built using attention weights and skip connections (Abnar & Zuidema, 2020). Suppose $w_{i,j}^{att}$ is the $i$-th row $j$-th column element of attention matrix averaged across heads. For self-attention, the edge weight $e_{i,j}$ between any two tokens, $i$ and $j$, is $e_{i,j} = w_{i,j}^{att} + 1$ if $i = j$, indicating a skip-connection, and $e_{i,j} = w_{i,j}^{att}$ if $i \neq j$. In the case of cross-attention, edge $e_{i,j} = w_{i,j}^{att}$ connects text to video, and $e_{i,i} = 1$ for connections within video tokens due to skip connections. Given that connections only exist from one layer to the next, the resulting matrix exhibits block-wise sparsity pattern. This is expressed as $\boldsymbol{W} = \begin{bmatrix} \mathbf{0} & E_{t,1} & \mathbf{0} & \dots & E_{t,l-1} & \mathbf{0} \\ \mathbf{0} & \mathbf{0} & E_2 & \dots & \mathbf{0} & \mathbf{0} \\ \mathbf{0} & \mathbf{0} & \mathbf{0} & \dots & \vdots & \vdots \\ \mathbf{0} & \mathbf{0} & \mathbf{0} & \dots & \mathbf{0} & E_l \end{bmatrix}$.

Here, $\boldsymbol{W}$ is a block matrix composed of smaller matrices $E_l$ and $E_{t,l}$. Each element within $E_l$ and $E_{t,l}$ represents the edge weight between two tokens. Specifically, $E_l$ denotes the edge weights within video tokens at $l$-th layer, and $E_{t,l}$ indicates the influence from text to video at $l$-th cross-attention layer. In this structure, the text tokens correspond to the first row and first column of $\boldsymbol{W}$, while the video tokens are represented by the remaining rows and columns. The remaining values are set to 0, because there are no direct connections between tokens from different layers.

**Attribution as Flow on Graph.** Given graph $G$, we compute the attribution $A_i$ by analyzing all paths from a text token $v_i$ to video tokens at the output layer. As such, we formulate it as a *max-flow problem* with capacity matrix $\boldsymbol{W}$. To facilitate this, we add an auxiliary target node $\boldsymbol{v}_t$ to $G$, connecting it to all output video tokens with inflow edges $e_{\boldsymbol{v}_t^+} = 1$[1]. We treat each text token $v_i$ as the source, and $\boldsymbol{v}_t$ as the sink. The max-flow from source to sink quantifies the influence of $v_i$, termed *ST-Flow*.

**Definition 1** (ST-Flow). *In attention graph $G$ with capacity matrix $\boldsymbol{W}$, a input token $v_i$ as source and sink node $\boldsymbol{v}_t$, the attribution value of $A_i = |f|^*$ is computed as the maximum flow from $v_i$ to $\boldsymbol{v}_t$.*

Our ST-Flow can be considered as an extension of Attention Flow (Abnar & Zuidema, 2020), incorporating all attention layers in diffusion model. It is proved to be a kind of Shapley Value (Ethayarajh & Jurafsky, 2021), which is an ideal contribution allocation in game theory (Shapley et al., 1953;

---

[1]The maximum inflow is 1 for each node due to softmax normalization in the attention.

Myerson, 1977; Young, 1985) and interpretable AI model (Lundberg & Lee, 2017b; Sundararajan et al., 2017).

**Exact ST-Flow Computation is infeasible.** While theoretically possible, calculating the ST-Flow in T2V diffusion models faces practical issues that render it infeasible:

- **Non-Differentiable.** The max-flow algorithm, by its nature, is non-differentiable. This is a problem when we do gradient-based optimization in Eq 4.

- **Efficiency Issue.** Solving max-flow for each input token is slow. Even with the Dinic's algorithm (Dinic, 1970) [2], the time complexity is $O(K|V|^2|E|)$ for large attention graphs in video.

Despite these obstacles, in Sec 3.3, we derive a min-max approximation to circumvent these issues.

### 3.3 Differentiable ST-Flow with Min-Max Path Flow

As discussed above, exact computation of ST-Flow is challenging. Instead of directly estimating the ST-Flow, we approach this by focusing on approximating its lower bound, which is computationally feasible. This is made possible, since any sub-graph has max-flow smaller than that of full graph.

**Theorem 1** (Sub-Graph Flow)[3]. *For any sub-graph $g$ of a graph $G$, $g \subseteq G$, the maximum flow $f_g^*$ in $g$ is less than or equal to the maximum flow $f_G^*$ in $G$, $|f_g^*| \leq |f_G^*|$.*

Based on this theorem, we need not compute the ST-Flow directly. Instead, we sample multiple subgraphs $g$ from $G$, calculate the maximum flow for each, and take the highest value among these:

$$|f_G| \geq A_i = \max_{\forall g \subseteq G} |f_g|; \tag{5}$$

This approach allows for a more efficient calculation by focusing on a manageable number of subgraphs, solving the max-flow for each, and identifying the maximum flow.

In this work, we focus on the simplest type of subgraph in graph $G$: a path from a $v_i$ to target $\boldsymbol{v}_t$. We efficiently approximate the ST-Flow by computing the *max path flow* for each path. We propose two min-max strategies to achieve this:

- **Hard Flow Strategy.** For each text token $v$, we sample all paths $v_i$ to $v_t$. The max-flow on each path is calculated as the minimum edge capacity along the path, $|f| = \min_j e_j$. And the best approximated $A_i = \max |f|$ is the maximum of these minimums across all paths.

- **Soft Flow Strategy.** Instead of get the hard min-max flow, we use *soft-min* and *soft-max* operations using the log-sum-exp trick. This approach provides a smoother approximation of flow values, which can be especially useful in our gradients-based optimization. The soft-min/max is computed as below, with $\tau$ as a temperature

$$\text{softmax}(e_1, e_2, \ldots; \tau) = \tau \log \left( \sum_j \exp \left( \frac{e_j}{\tau} \right) \right); \tag{6}$$

$$\text{softmin}(e_1, e_2, \ldots; \tau) = -\text{softmax}(-e_1, -e_2, \ldots; \tau), \tag{7}$$

**Vectorized Path Flow Computation.** While depth-first and breadth-first searches can identify all paths for above min-max optimization, these methods are slow and cannot be parallelized. Instead, we define a special operation called *min-max multiplication* on the capacity matrix to calculate the maximum flow for each path in a vectorized manner.

**Definition 2** (Min-max Multiplication). *Given two matrices $A \in \mathbb{R}^{m \times k}$ and $B \in \mathbb{R}^{k \times n}$, min-max multiplication $C = A \odot B \in \mathbb{R}^{m \times n}$ is defined where each element $C_{i,j} = \max_r(\min(A_{i,r}, B_{r,j}))$.*

This operation computes the minimum value across all $r$ for the $i$-th row of $A$ and the $j$-th column of $B$, and $\max_r$ selects the maximum of these minimum values for each $C_{i,j}$. We call it a *multiplication* because it resembles matrix multiplication but replaces element-wise multiplication with a minimum operation and summation with maximization.

---

[2] Given that the attentions has more edge than tokens, Dinic is best choice in theory. However, our implementation shows that max-flow on each token takes ~8s.

[3] Proof in Appendix A

A very good property is that, the min-max multiplication of capacity matrix $W^k = W^{k-1} \odot W$ can be interpreted as the max path flow for all $k$-hop paths.

**Proposition 1** (Max Path Flow using Min-max Multiplication)[4]. *For min-max power of capacity $W^k = W^{k-1} \odot W$, element $W_{i,j}^k$ equals the max path flow for all $k$-hop path from $v_i$ to $v_j$.*

For attention graph that current layer's node is only connect to the next layer, all path from text token to output video token has exactly the length of $l$. In this way, what we do is just to extract the attention graph $G$, do $l$ times Min-max Multiplication on its flow matrix, and we consider the value as a approximation of ST-Flow. A tine complexity analysis is prepared in Appendix G.

In this way, we get all pieces to build **Vico**. We first compute attribution using the approximated ST-Flow, then using Eq 4 to update the latent to equalize such flow.

## 4 EXPERIMENTS

In our experiments section, we evaluate **Vico** through a series of tests. We start by assessing its performance on generating videos from compositional text prompts. Next, we demonstrate ST-Flow accurately attributes token influence through video segmentation and human study. We also conduct an ablation study to validate our key designs. More application results are provide in Appendix E and Appendix D.

### 4.1 EXPERIMENT SETUP

**Baselines.** We build our method on several open-sourced video diffusion model, including VideoCrafterv2 (Chen et al., 2024), AnimateDiff (Guo et al., 2024) and Zeroscopev2 [5]. Since no current compositional generation method are specifically designed for video, we re-implement several methods designed for text-to-image diffusion models and compare with them. These methods include:

- *Original Model*. We directly ask the original base model to produce video based on prompts.

- *Token Re-weight*. We use the `compel` [6] package to directly up-lift the weight of specific concept token, with a fixed weight of 1.5.

- *Compositional Diffusion* (Liu et al., 2022). This method directly make multiple noise predictions on different text, and sum the noise prediction as the compositional direction for latent update. In our paper, given a prompt, we first split into short phrases. For example "`a dog and a cat`" is splitted into "`a dog`" and "`a cat`", make individual denoising, and added up.

- *Attend-and-Excite* (Chefer et al., 2023). A&E refines the noisy latents to excite cross-attention units to attend to all subject tokens in the text prompt.

Besides those training-free methods, we also includes some recent work that retrain the diffusion model for compositional generation. These includes LVD (Lian et al., 2023) and VideoTetris (Tian et al., 2024).

**Evaluation and Metrics.** We evaluate compositional generation using VBench (Huang et al., 2024) and T2V-CompBench(Sun et al., 2024). Specifically, we focus on evaluating compositional quality in terms of *Spatial Relation*, *Multiple Object Composition*. For both metrics, the model processes text containing multiple concepts, generates a video. Then a caption model verifies the accuracy of the concept representations within the generated video.

Additionally, we design a new metric, *Motion Composition*. This metric evaluates the generated video based on the presence and accuracy of multiple objects performing different motions. We collect 70 prompts of the form "`obj`$_1$ `is motion`$_1$ `and obj`$_2$ `is motion`$_2$". Using GRiT (Wu et al., 2022), we generate dense captions on video for each object and verify if each (`object, motion`) pair

---

[4]Proof in Appendix B

[5]https://huggingface.co/cerspense/zeroscope_v2_576w

[6]https://github.com/damian0815/compel

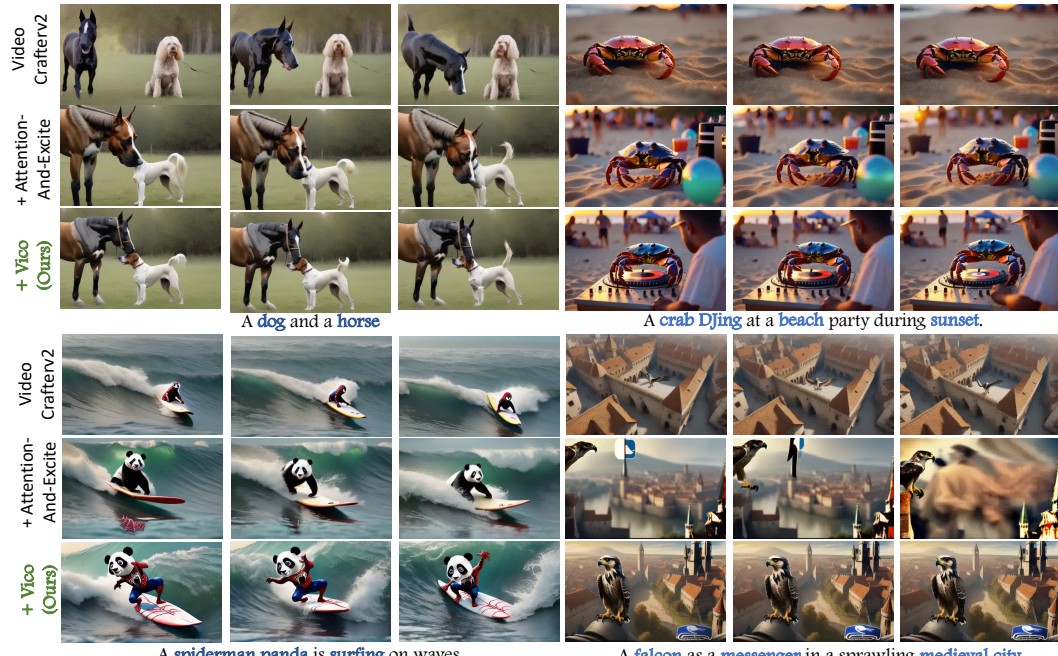

Figure 4: Qualitative comparison of the videos generated by VideoCrafterv2 baseline, Attribute&Excite and our **Vico** with compositional textual descriptions.

appears in the captions. The score is computed as $\frac{\sum_{1,2}(\mathbb{I}(\text{obj}_i)+\mathbb{I}(\text{obj}_i,\text{motion}_i))}{4}$. Here, $\mathbb{I}(x)$ is an indicator function that returns 1 if $x$ is present in the generated captions, and 0 otherwise.

The overall video quality is measured using ViCLIP (Wang et al., 2023c) to compute a score based on text and video alignment, denoted as *Overall Consistency*.

We also report the 5 metrics in T2V-CompBench, including *Consistent-Attribute Bidding*, *Spatial Relations*, *Motion Bidding*, *Action Bidding* and *Object Interations*.

**Implementation Details.** We use the implementations on `diffusers` for video generation. All videos are generated by a A6000 GPU. We sample videos from Zeroscopev2 and VideoCrafterv2 using 50-step DPM-Solver++ (Lu et al., 2022). AnimateDiff is sampled with 50-step DDIM (Song et al., 2020). We optimize the latent at each sampling steps, and update the latent with Adam (Kingma & Ba, 2014) optimizer at the learning rate of $1e-5$. We test both the soft and hard-min/max versions of Vico, setting the temperature $\tau = 0.01$ for the soft version. The NLTK package identify all nouns and verbs for equalization.

### 4.2 COMPOSITIONAL VIDEO GENERATION

**Quantitative Results.** In Table 1, we present the scores achieved by **Vico** compared to other methods across various base models on compositional text-to-video generation. Vico consistently surpasses all baselines on every metric. Notably, our ST-flow based method surpasses cross-attention based techniques like Attend&Excite, thanks to its ability to incorporating influences across full attention graph. Additionally, the soft min-max version of Vico generally achieves better fidelity than the hard version, as it is better suited for gradient optimization.

Surprisingly, Vico demonstrates its most significant improvements in multi-subject generation tasks. For instance, on VideoCrafter2, it shows a marked increase, improving scores from $40.66\% \rightarrow 73.55\%$. This suggests that our attention mechanism in T2V is more adept at managing object arrangement. In contrast, compositional diffusion models often fail, as they assume conditions to be independent, which is problematic for complex compositions.

Table 1: Quantitative results for different methods on compositional text-to-video generation.

| Name | Spatial Relation↑ | Multiple Object↑ | Motion Composition↑ | Overall Consistency↑ |
|---|---|---|---|---|
| AnimateDiff (Guo et al., 2024) | 24.80% | 33.44% | 33.90% | 27.75% |
| +Compositional Diffusion (Liu et al., 2022) | 19.43% | 7.27% | 23.58% | 24.07% |
| +Attend-and-Excite (Chefer et al., 2023) | 20.88% | 31.25% | 34.78% | 28.05% |
| +Token-Reweight | 28.11% | 36.89% | 37.45% | 26.77% |
| +`Vico` (*hard*) | 24.22% | 29.95% | 37.23% | 28.85% |
| +`Vico` (*soft*) | **31.47**% | **37.20**% | **37.95**% | **28.89**% |
| ZeroScopev2 | 59.52% | 52.52% | 45.51% | 25.83% |
| +Compositional Diffusion (Liu et al., 2022) | 31.77% | 8.23% | 33.13% | 23.02% |
| +Token-Reweight | 57.48% | 50.00% | 40.42% | 25.74% |
| +Attend-and-Excite (Chefer et al., 2023) | 59.02% | 62.27% | 45.82% | 25.84% |
| +`Vico` (*hard*) | **63.60**% | 63.34% | **46.32**% | 24.89% |
| +`Vico` (*soft*) | 62.28% | **69.05**% | 45.31% | **26.15**% |
| VideoCrafterv2 (Chen et al., 2023) | 35.86% | 40.66% | 43.82% | 28.06% |
| +Compositional Diffusion (Liu et al., 2022) | 23.61% | 10.59% | 35.49% | 24.49% |
| +Token-Reweight | 46.08% | 49.16% | 44.33% | 28.29% |
| +Attend-and-Excite (Chefer et al., 2023) | 48.11% | 66.62% | 43.48% | 28.33% |
| +`Vico` (*hard*) | 49.85% | 67.84% | 44.46% | 28.41% |
| +`Vico` (*soft*) | **50.40**% | **73.55**% | **44.98**% | **28.52**% |

Table 2: Comparison of Models on T2V CompBench.

| Model | Consist-attr | Spatial | Motion | Action | Interaction |
|---|---|---|---|---|---|
| LVD (Lian et al., 2023) | 0.5595 | **0.5469** | **0.2699** | 0.4960 | 0.6100 |
| VideoTetris (Tian et al., 2024) | **0.7125** | 0.5148 | 0.2204 | 0.5280 | 0.7600 |
| VideoCrafterv2+`Vico` (*soft*) | 0.6980 | 0.5432 | 0.2412 | **0.6020** | **0.7800** |

In addition, we compare our method combined with VideoCrafterv2 against advanced video diffusion advanced video diffusion models like LVD (Lian et al., 2023) and VideoTetris (Tian et al., 2024). These models use bounding box supervision or curated datasets. The results in Table 2 show that our method performs similarly. It even outperforms on *action binding* and *object interactions*, without relying on external data or additional training.

**Qualitative Results.** We compare the videos generated by different methods in Figure 4. Attend&Excite receive slightly improvements, but still mixes semantics of different subject. For example, on the "`a dog and a horse`" example (Top Left), both Attend&Excite and the baseline incorrectly combine a dog's face with a horse's body. Vico addresses this issue by ensuring each token contributes equally, effectively separating their relationships.

Additionally, cross-attention often leads to temporal inconsistencies in the modified videos. For instance, in the "`spider panda`" case (Bottom Left), Attend&Excite initially displays a Spider-Man logo but it disappears abruptly in subsequent frames. In contrast, Vico captures dynamics across both spatial and temporal attention, leading to better results. More results is in Appendix D and E.

## 4.3 ATTRIBUTION ON VIDEO DIFFUSION MODEL

In this section, we aim to demonstrate that our ST-Flow (hard) provides a more accurate measure of token contribution compared to other attention-based indicators.

**Objective Evaluation: Zero-shot Video Segmentation.** We tested several attribution methods using the VideoCrafterv2 model for zero-shot video segmentation on the Ref-DAVIS2017 (Khoreva et al., 2019) dataset. To create these maps, we first performed a 25-step DDIM inversion (Mokady et al., 2023) to extract noise patterns, followed by sampling to generate the attribution maps. We specifically used maps from from *end of text* ([EOT]) token (Li et al., 2024) for segmentation. We used the mean value of the map as a threshold for binary classification. We compare with cross-attention (Tang et al., 2023) and Attention Rollout (Abnar & Zuidema, 2020). The more accurate the segmentation is, the attribution is more reasonable for human.

| Attribution Method | Temporal Consistency↑ | Reasonability↑ |
|---|---|---|
| Cross-Attention | 2.62±0.12 | 2.87±0.23 |
| Attention Rollout | 3.77±0.20 | 3.36±0.20 |
| ST-Flow (Ours) | **4.12**±0.13 | **3.76**±0.19 |

Table 3: User study on attribution method.

| Method | Ref-DAVID2017 | | |
|---|---|---|---|
| | $\mathcal{J}\&\mathcal{F}\uparrow$ | $\mathcal{J}\uparrow$ | $\mathcal{F}\uparrow$ |
| Supervised Trained | | | |
| ReferFormer-B | 61.1 | 58.1 | 64.1 |
| OnlineRefer-B | 62.4 | 59.1 | 65.6 |
| Zero-Shot | | | |
| Cross-Attention mean | 32.1 | 29.8 | 34.7 |
| Attention Rollout mean | 38.0 | 33.3 | 40.0 |
| ST-Flow (Ours) mean | **38.2** | **33.5** | **40.3** |

Table 4: Performance on Ref-DAVID2017.

| Min Loss | ST-Flow (soft) | Multiple Object↑ | Overall Consistency↑ |
|---|---|---|---|
| ✗ | ✗ | 57.86% | 28.03% |
| ✓ | ✗ | 63.62% | 28.24% |
| ✗ | ✓ | 69.75% | 28.12% |
| ✓ | ✓ | **73.55%** | **28.52%** |

Table 5: Ablation study on Vico.

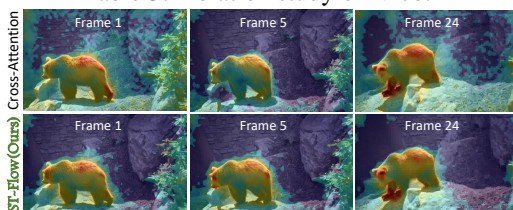

Table 6: Segmentation results comparison.

Results are presented in Table 4. Our method outperformed the others, providing the highest segmentation metrics in zero-shot setting. As visualized in Figure 6, cross-attention maps showed inconsistent highlighting and flickering. Attention Rollout also concider the full attention graph, but overly smoothed weights, resulting in less precise object focus.

**Subjective Evaluation: User Study.** Besides, segmentation-based validation, we conducted a subjective user study to evaluate the quality of attribution maps generated by various methods. 43 participants rated maps from three different approaches across 50 video clips. The evaluation focused on *Temporal Consistency*, assessing the presence of flickering, and *Reasonability*, determining alignment with human interpretations. Ratings ranged from 1 to 5, with 5 as the highest. As summarized in Table 3, Our ST-Flow method outperformed others, achieving the highest scores in both Temporal Consistency (4.12) and Reasonability (3.76).

### 4.4 ABLATION STUDY

In this subsection, we ablate our two key designs: the loss function and the proposed ST-Flow.

**Loss Function.** We modified the loss function from using the "min" as a fairness indicator (as described in Sec 3.1) to a variance loss, defined as $\mathcal{L}_{\text{fair}} = -\sum_i (A_i - \bar{A})^2$. This aims to minimize the differences between each $A_i$ and the average attribution value $\bar{A}$, making it fair. The results is shown in Table 5, row 3 and 4. We notice while the variance loss ensures uniformity across all tokens, it overly restricts them, often degrading video quality. Conversely, our original min-loss focuses on the least represented token, enhancing object composition accuracy without significantly affecting overall quality.

**ST-Flow** *v.s.* **Cross-Attention.** A major contribution of our work is the development of ST-Flow and its efficient computation. We compared it against a model using cross-attention, where cross-attention maps are extracted and a mean score is computed for each token as $A_i$. As demotivated in Table 5, row 2 and 4, using ST-Flow (soft) largely outperform cross-attention. We also provide the running speed analysis in Appendix G, confirming the efficiency of our approach.

### 5 CONCLUSION

In this paper, we present **Vico**, a framework designed for compositional video generation. Vico starts by analyzing how input tokens influence the generated video. It then adjusts the model to ensure that no single concept dominates. To implement Vico practically, we calculate each text token's contribution to the video token using max flow. This computation is made feasible by approximating the subgraph flow with a vectorized implementation. We have applied our method across various diffusion-based video models, which has enhanced both the visual fidelity and semantic accuracy of the generated videos.

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

## A    PROOF OF THEOREM 1: SUB-GRAPH FLOW

In a network $G = (V, E)$ with a capacity function $c : E \to \mathbb{R}^+$, and a subgraph $g$ of $G$, the maximum flow $f_g$ in $g$ is less than or equal to the maximum flow $f_G$ in $G$.

PROOF

1. **Definition of a Subgraph:** A subgraph $g$ of $G$ can be defined as $g = (V', E')$ where $V' \subseteq V$ and $E' \subseteq E$. All capacities in $g$ are inherited from $G$, i.e., $c'(e) = c(e)$ for all $e \in E'$.

2. **Flow Conservation:** Both $G$ and $g$ must satisfy the flow conservation law at all intermediate nodes. That is, the sum of the flow entering any node must equal the sum of the flow exiting that node, except for the source (where flow is generated) and the sink (where flow is absorbed).

3. **Reduced Set of Paths:** Since $E' \subseteq E$, every path through $g$ is also a path through $G$, but not every path through $G$ is necessarily a path through $g$. This reduction in the number of paths (or edges) in $g$ implies that some routes available for flow in $G$ are not available in $g$.

4. **Capacity Limitations:** For any edge $e$ in $E'$, the capacity in $g$ (i.e., $c'(e)$) equals the capacity in $G$ (i.e., $c(e)$). Therefore, no edge in $g$ can support more flow than it can in $G$. Additionally, since some edges might be missing in $g$, the overall capacity of pathways from the source to the sink might be less in $g$ than in $G$.

5. **Maximum Flow Reduction:** Given the reduction in paths and capacities, any flow that is feasible in $g$ is also feasible in $G$, but not vice versa. Hence, the maximum flow $f_g$ that can be pushed from the source to the sink in $g$ must be less than or equal to the maximum flow $f_G$ that can be pushed in $G$.

**Conclusion:** From these points, it follows directly that the maximum flow in a subgraph $g$ cannot exceed the maximum flow in the original graph $G$. This proves that $f_g \leq f_G$.

## B    PROOF OF PROPOSITION 1: MAX PATH FLOW USING MIN-MAX MULTIPLICATION

**Definitions and Proposition:** Let $\mathbf{W}$ be a capacity matrix of a graph where $\mathbf{W}_{i,j}$ is the capacity of the edge from vertex $i$ to vertex $j$. If there is no edge between $i$ and $j$, $\mathbf{W}_{i,j} = 0$ or some representation of non-connectivity. A $k$-hop path between two vertices $i$ and $j$ is a path that uses exactly $k$ edges.

**Proposition:** The $k$-th min-max power of $\mathbf{W}$, denoted $\mathbf{W}^k$, calculated as $\mathbf{W}^k = \mathbf{W}^{k-1} \odot \mathbf{W}$, has elements $\mathbf{W}^k_{i,j}$ that represent the maximum flow possible on any $k$-hop path from vertex $i$ to $j$.

**Min-max Multiplication:** Given matrices $\mathbf{A}$ and $\mathbf{B}$, $\mathbf{C} = \mathbf{A} \odot \mathbf{B}$ is defined such that:

$$\mathbf{C}_{i,j} = \max_r (\min(\mathbf{A}_{i,r}, \mathbf{B}_{r,j}))$$

**Proof by Induction:**

**Base Case** ($k = 1$):

- **Claim:** $\mathbf{W}^1_{i,j}$ represents the capacity of the edge from $i$ to $j$, which is the maximum flow on a 1-hop path.
- **Proof:** By definition, $\mathbf{W}^1 = \mathbf{W}$, and $\mathbf{W}^1_{i,j} = \mathbf{W}_{i,j}$, which directly corresponds to the edge capacity between $i$ and $j$. Hence, the base case holds.

**Inductive Step:**

- **Assumption:** Assume that for $k - 1$, $\mathbf{W}^{k-1}_{i,j}$ correctly represents the maximum flow on any $k - 1$-hop path from $i$ to $j$.

- **To Prove:** $\mathbf{W}_{i,j}^k$ represents the maximum flow on any $k$-hop path from $i$ to $j$.

**Proof:** From the definition of min-max multiplication,

$$\mathbf{W}_{i,j}^k = \max_r(\min(\mathbf{W}_{i,r}^{k-1}, \mathbf{W}_{r,j}))$$

- $\mathbf{W}_{i,r}^{k-1}$ is the maximum flow from $i$ to $r$ using $k-1$ hops.
- $\mathbf{W}_{r,j}$ is the capacity of the edge from $r$ to $j$ (1-hop).

**Interpretation:** $\min(\mathbf{W}_{i,r}^{k-1}, \mathbf{W}_{r,j})$ finds the bottleneck flow for the path from $i$ to $j$ through $r$ using $k$ hops. The minimum function ensures the path's flow is constrained by its weakest segment.

**Maximization Step:** $\max_r$ over all possible intermediate vertices $r$ selects the path with the highest bottleneck value, thus ensuring the selected path is the most capable among all possible $k$-hop paths.

**Conclusion:** The inductive step confirms that the flow represented by $\mathbf{W}_{i,j}^k$ is indeed the maximum possible flow across any $k$-hop path from $i$ to $j$. Hence, by induction, the proposition holds for all $k$.

## C  RELATED WORK

**Video Diffusion Models.** Video diffusion models generate video frames by gradually denoising a noisy latent space (Ho et al., 2022b). One of the main challenges with these models is their high computational complexity. Typically, the denoising process is performed in the latent space (Zhou et al., 2022; Blattmann et al., 2023b;a). The architectural commonly adopt either a 3D-UNet (Ho et al., 2022b; Blattmann et al., 2023b; Ho et al., 2022a; Harvey et al., 2022; Wu et al., 2023a) or diffusion transformer (Gupta et al., 2023; Peebles & Xie, 2023; Ma et al., 2024). To enhance computational efficiency, these architectures often employ separate self-attention mechanisms for managing spatial and temporal tokens. Conventionally, training these models involves fine-tuning an image-based model for video data (Wu et al., 2023a; Khachatryan et al., 2023; Guo et al., 2024). This process includes adding a temporal module while striving to preserve the original visual quality.

Despite their ability to generate photorealistic videos, these models frequently struggle with understanding the complex interactions between elements in a scene. This shortcoming can result in the generation of nonsensical videos when responding to complex prompts.

**Compositional Generation.** Current generative models often face challenges in creating data from a combination of conditions, with most developments primarily in the image domain. Energy-based models (Du et al., 2020; 2023; Liu et al., 2023), for example, are mathematically inclined to be compositionally friendly, yet they require the conditions to be independent. In practice, many image-based methods utilize cross-attention to effectively manage the composition of concepts (Feng et al., 2023; Chefer et al., 2023; Wu et al., 2023b; Rassin et al., 2024). However, when it comes to video, compositional generation introduces additional complexities. Some video-focused approaches concentrate specific form of composition, including object-motion composition (Wei et al., 2023), subject-composition (Wang et al., 2024b), utilize explicit graphs to control content elements (Bar et al., 2021). Others incorporate multi-modal conditions (Wang et al., 2024a), additional training data (Tian et al., 2024), or auxiliary modules (Lian et al., 2023). Despite these efforts, a generic solution for accurately generating videos from text descriptions involving multiple concepts is still lacking. We present the first training-free solution for compositional video generation using complex text prompts, an area that remains largely under-explored.

**Attribution Methods.** Attribution methods clarify how specific input features influence a model's decisions. gradient-based methods (Sundararajan et al., 2017; Simonyan et al., 2013; Selvaraju et al., 2017) identify influential image regions by back-propagating gradients to the input. Attention-based methods (Chefer et al., 2021; Abnar & Zuidema, 2020) that utilize attention scores to emphasize important inputs. Ablation methods(Ramaswamy et al., 2020; Zeiler & Fergus, 2014) modify data parts to assess their impact. Shapley values (Lundberg & Lee, 2017a) distribute the contribution of each feature based on cooperative game theory. In our paper, we extend existing techniques of attention flow to video diffusion models. We develop an efficient approximation to solve the max-flow problem. This improvement helps us more accurately identify and balance the impact of each textual elements on synthesized video.

## D  COMPOSITIONAL VIDEO EDITING

Our system, Vico, can be integrated into video editing workflows to accommodate text prompts that describe a composition of concepts.

**Setup.** We begin by performing a 50-step DDIM inversion on the input video. Following this, we generate a new video based on the given prompt.

**Results.** An example of this process is illustrated in Figure 9. The original video demonstrates a strong bias towards a single presented object, making editing with a composition of concepts challenging. However, by applying Vico, we successfully enhance the video to accurately represent the intended compositional concepts.

## E  MORE VISUALIZATIONS

Here we provide more example for compositional T2V in Figure 5

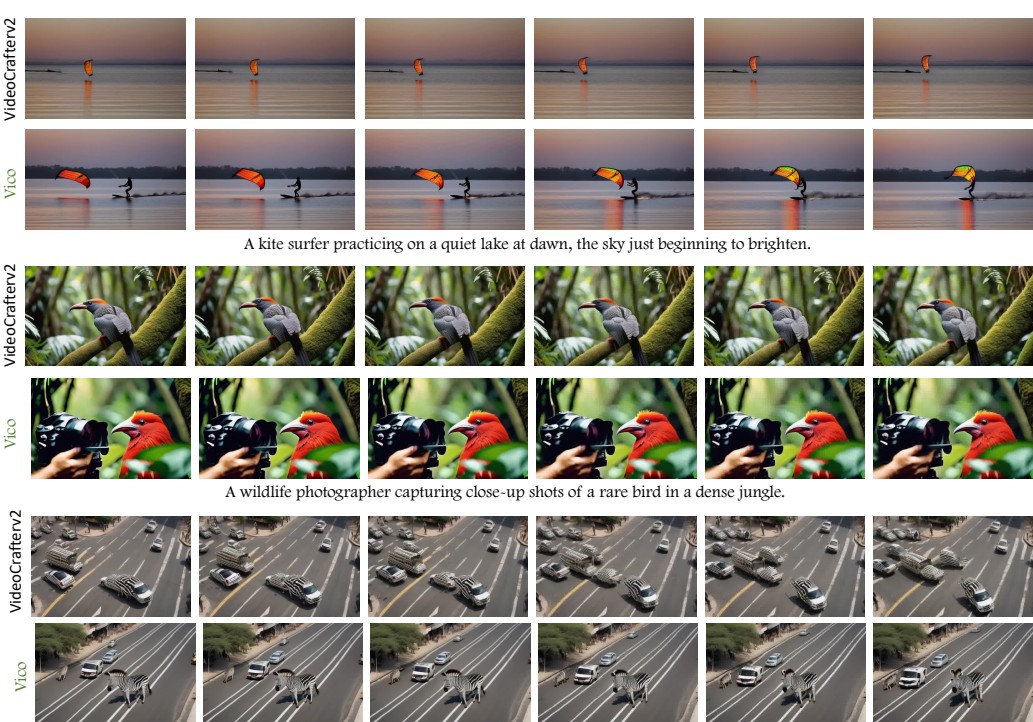

Figure 5: Video visualization for compositional video generation

### E.1  MOTION COMPOSITION

We visualize examples generated under motion composition scenarios, where the diffusion model is given text description that multiple objects exhibit distinct movement patterns. We compared results generated with VideoCrafterv2 to those produced by our method, Vico, using prompts from our motion composition evaluation.

The results are shown in Figure 8. Our method demonstrates clear improvements by effectively binding different actions to their respective subjects.

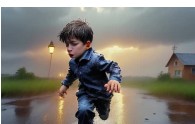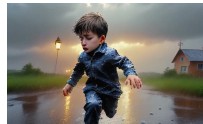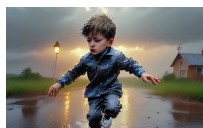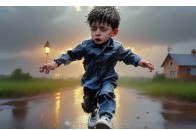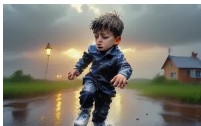

A small boy, head bowed and determination etched on his face, sprints through the torrential downpour as lightning crackles and thunder rumbles in the distance. The relentless rain pounds the ground, creating a chaotic dance of water droplets that mirror the dramatic sky's anger. In the far background, the silhouette of a cozy home beckons, a faint beacon of safety and warmth amidst the fierce weather. The scene is one of perseverance and the unyielding spirit of a child braving the elements.

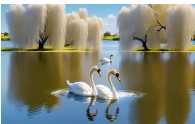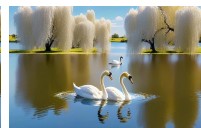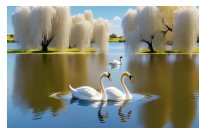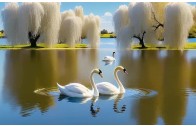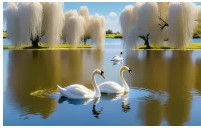

On a brilliant sunny day, the lakeshore is lined with an array of willow trees, their slender branches swaying gently in the soft breeze. The tranquil surface of the lake reflects the clear blue sky, while several elegant swans glide gracefully through the still water, leaving behind delicate ripples that disturb the mirror-like quality of the lake. The scene is one of serene beauty, with the willows' greenery providing a picturesque frame for the peaceful avian visitors.

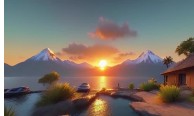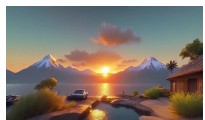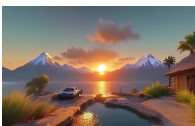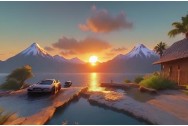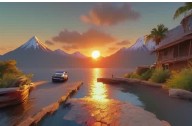

Create a visually stunning video that captures the journey of a lone traveler exploring diverse landscapes. Begin with a serene sunrise over a mountain range, transition to bustling city streets, and conclude with a tranquil seaside sunset. Incorporate dynamic camera movements, natural lighting, and rich textures to evoke a sense of adventure and serenity. Blend realistic visuals with a touch of artistic flair to create an engaging and emotive visual narrative.

Figure 6: Videos generated with long prompts.

### E.2 Long Prompt

We also demonstrate the capability of Vico to handle extremely long textual prompts. As shown in Figure 6, Vico effectively generates complex interactions between various concepts even with lengthy input prompts.

## F Adaptation of Vico to Diffusion Transformer Models

While our method is initally presented in UNet architecture, we can build it on recent video diffusion model with transformer. For example, we built Vico on top of Open-Sora (Zheng et al., 2024) and CogVideoX (Yang et al., 2024), adapting it to their respective architectures.

**Open-Sora Adaptation** . Open-Sora (Zheng et al., 2024) employs STDiT architecture, which separates spatial and temporal attention. This straightforward design made it relatively simple to adapt Vico for integration. By leveraging its design, we seamlessly incorporated Vico's token re-weighting mechanism into Open-Sora.

**CogVideoX Adaptation** . CogVideoX (Yang et al., 2024), in contrast, employs a more complex 3D MM-DiT architecture. It processes all text and video tokens jointly through a unified attention layer, without explicit cross-attention mechanisms. This design posed a unique challenge for traditional cross-attention control methods. However, Vico's graphical abstraction approach proved highly effective in this setting, as the model still fundamentally operates on token-to-token attention.

To adapt Vico to CogVideoX, we redefined the graph construction rules as follows:

$$\mathbf{W}_l = \begin{bmatrix} E_{tt,l} & E_{tv,l} \\ E_{vt,l} & E_{vv,l} \end{bmatrix},$$

$$\mathbf{W} = \begin{bmatrix} \mathbf{W}_1 & \mathbf{0} & \dots & \mathbf{0} \\ \mathbf{0} & \mathbf{W}_2 & \dots & \mathbf{0} \\ \vdots & \vdots & \ddots & \vdots \\ \mathbf{0} & \mathbf{0} & \dots & \mathbf{W}_L \end{bmatrix}.$$

Here, $\mathbf{W}_l$ represents the adjacency matrix at layer $l$, where $E_{tt,l}$, $E_{tv,l}$, $E_{vt,l}$, and $E_{vv,l}$ correspond to text-to-text, text-to-video, video-to-text, and video-to-video connections, respectively. Each $E$ is calculated as described in Line 236 of the main text. Stacking these matrices across all layers yields the final capacity matrix $\mathbf{W}$.

**Results on VBench** . We evaluated Vico with these adapted models on VBench, focusing on the *Multiple Object Composition* score. Due to the high memory requirements of MM-DiT, we used an 80GB A100 GPU for inference. The results, shown in Table 7, demonstrate that Vico significantly enhances performance across different architectures.

We also visualize several videos generated by CogVideoX using Vico in Figure 7. Even with modern video diffusion models like CogVideoX, compositional errors are still apparent. For instance, it blends *a boat and an airplane* into a single object, such as *a seaplane*, or generates only *a pizza* while neglecting *a tie*.

In contrast, Vico effectively resolves these conflicting objects and represents all concepts more accurately and fairly.

| Method | Multiple Object Composition Score |
|---|---|
| Open-Sora | 33.64 |
| Open-Sora + `Vico` | **48.21** |
| CogVideoX 2B | 53.70 |
| CogVideoX 2B + `Vico` | **63.21** |

Table 7: Performance comparison on VBench.

## G  SPEED ANALYSIS

**Attribution Speed.** In this section, we assess the running speed of our ST-flow. To assess its computational efficiency, we compare ST-flow with cross-attention and Attention Rollout (Abnar & Zuidema, 2020) computation, by reporting the theoretical complexity and empirical running time. We assume we have 1 cross attention map of $m \times n$ and $L$ self-attention map of $n \times n$, and demonstrated the theoretical results. Specifically, we measure the average running time required for each diffusion model inference, focusing solely on the time taken for attribution computation, excluding the overall model inference time. We use the VideoCrafterv2 as the base model.

As detailed in Table 8, the cross-attention computation is fast, as it processes only a single layer. Both Attention Rollout and our approximated ST-Flow involve matrix multiplications and consequently share a similar time complexity. However, our ST-Flow approximation benefits from the relatively faster speed of element-wise min-max operations compared to the floating-point multiplications used in Attention Rollout, leading to slightly quicker execution times.

In contrast, the exact ST-Flow method is much slower. This is because it requires independently estimating the flow for each sink-source pair, a process that takes considerable time.

**Diffusion Inference Speed.** Our Vico framework includes a iterative optimization process alongside with the denoising. As expected, it should results in longer inference time. We evaluated this using a 50-step DPM denoising process on the VideoCrafterv2 model, at a resolution of $512 \times 320$ for 16 frames, on a single A6000 GPU.

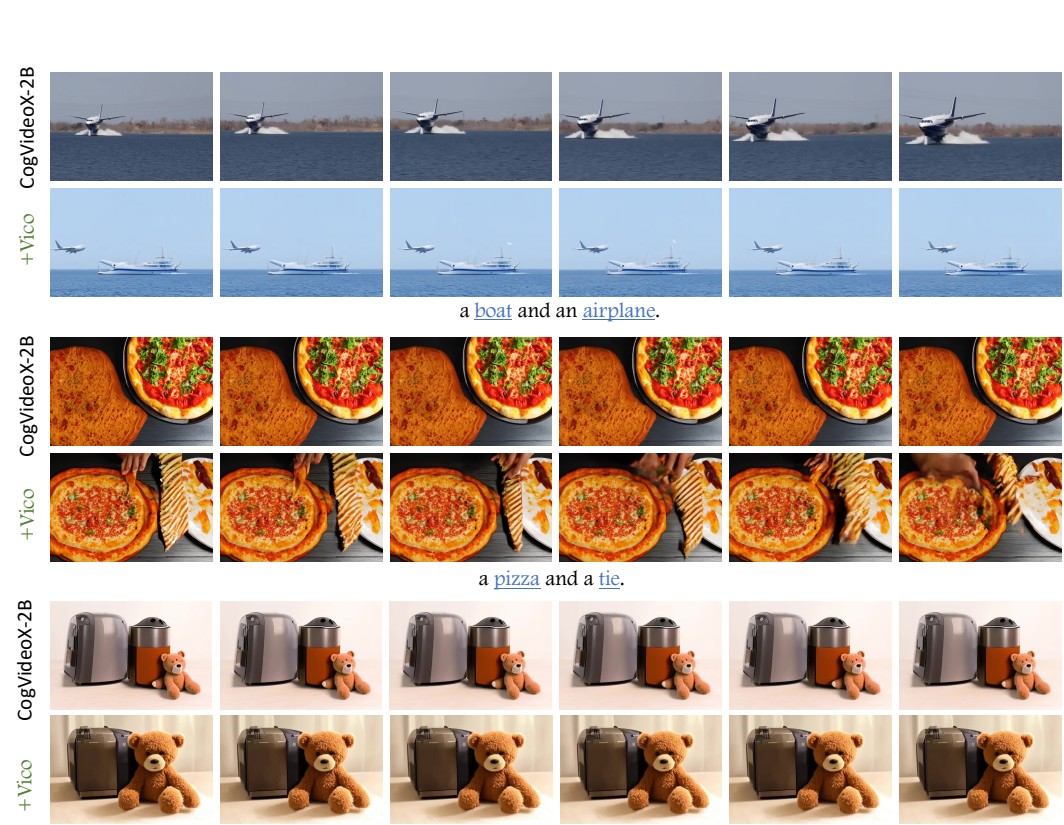

Figure 7: Compositional generation results on CogVideoX-2B.

| Method | Complexity | sec/inference |
|---|---|---|
| Cross-Attn. | $O(1)$ | 0.002s |
| Attention Rollout | $O(Lmn^2)$ | 0.042s |
| Exact-ST-Flow | $O(L^3mn^4)$ | 8s |
| ST-Flow (soft) | $O(Lmn^2)$ | 0.037s |

Table 8: Speed comparison for attribution method.

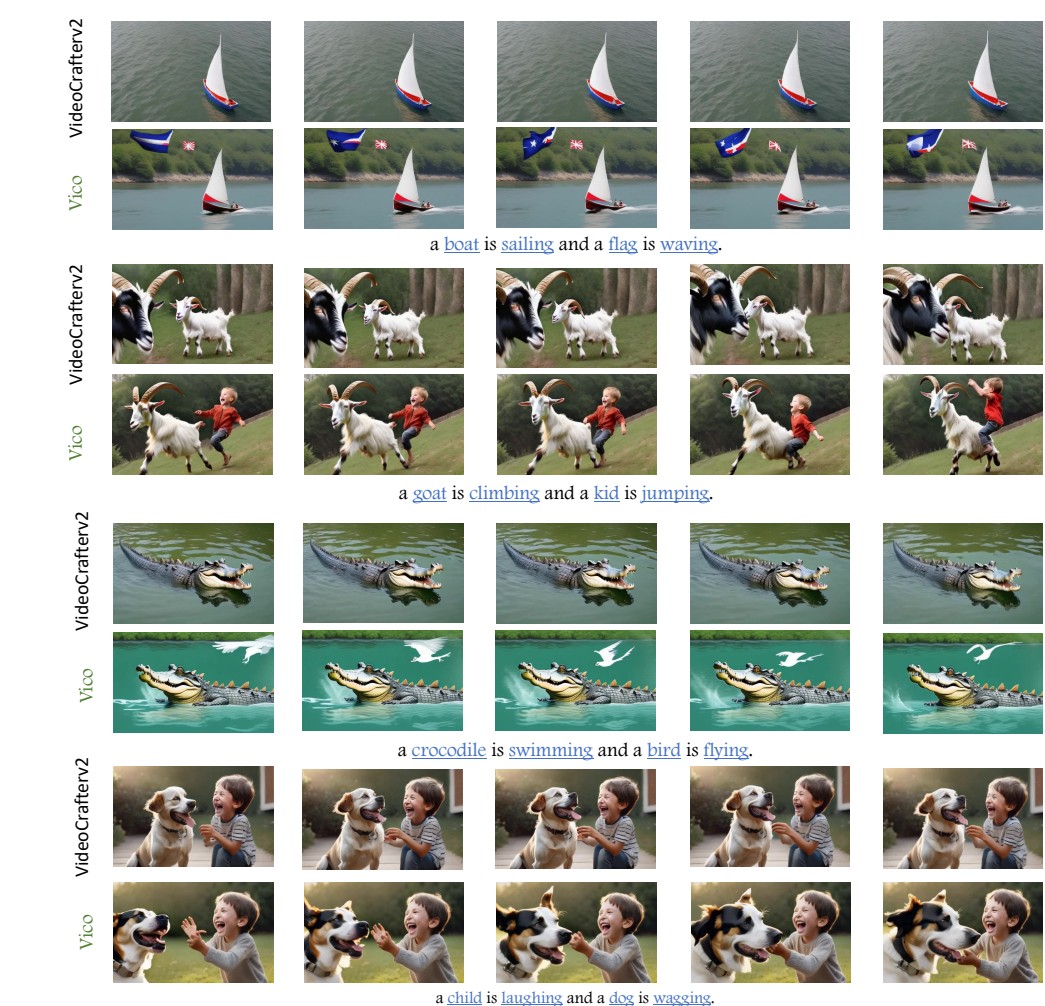

Figure 8: Video visualizations for prompts with motion composition.

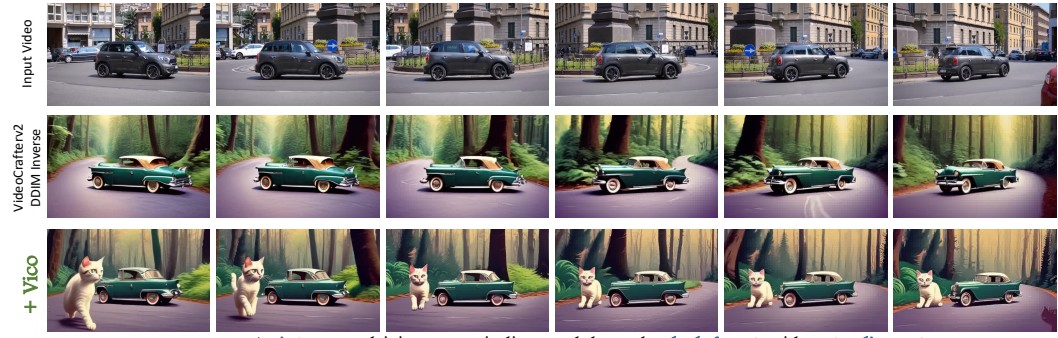

Figure 9: Video edit results with compositional prompts.

The results, shown in Table 9, reveal that the baseline VideoCrafterv2 completed in 23 seconds. Adding the Attend&Excite increased the duration to 48 seconds. In comparison, our Vico framework

| Method | Time |
|---|---|
| VideoCrafterv2 | 23s |
| + Attend&Excite | 48s |
| + `Vico` (soft&hard) | 45s |

Table 9: Text-to video model inference time comparison.

finished in a comparable time of 50 seconds. Despite its additional complexity, Vico's efficient design keeps the inference time within a reasonable range.

## H IMPLEMENTATION DETAILS OF VICO

**ST-Flow Computation.** To compute the ST-Flow, we begin by extracting attention weights from all layers. These weights are averaged across all heads and then upscaled to the image size using bicubic interpolation. Due to the block-wise sparse pattern of the connections, min-max matrix multiplication is applied to the capacity matrix for connected layers. Furthermore, given that cross-attention layers include skip connections from previous layers, we divide the network into multiple groups. Within each group, min-max matrix Multiplication is performed. Finally, we aggregate the scores across all groups to obtain the results. The pseudocode for the min-max multiplication is in Algorithm 1.

---

**Algorithm 1** Batched Min-Max Matrix Multiplication

---

1: **function** BATCHMINMAXMATRIXMULTIPLICATION($A, B$)
2:     **Input:**
3:     $A$ is a tensor of shape $[B, m, k]$
4:     $B$ is a tensor of shape $[B, k, n]$
5:     **Output:**
6:     Tensor of shape $[B, m, n]$ containing the maximum values

7:     $A_{\text{expanded}} \leftarrow A.\text{unsqueeze}(2)$          $\triangleright$ Shape becomes $[B, m, 1, k]$
8:     $B_{\text{expanded}} \leftarrow B.\text{permute}(0, 2, 1).\text{unsqueeze}(1)$     $\triangleright$ Shape becomes $[B, 1, n, k]$

9:     $min\_vals \leftarrow \text{torch.min}(A_{\text{expanded}}, B_{\text{expanded}})$     $\triangleright$ Shape becomes $[B, m, n, k]$
10:    $max\_vals \leftarrow \text{torch.max}(min\_vals, \dim = 3).\text{values}$     $\triangleright$ Shape becomes $[B, m, n]$

11:    **return** $max\_vals$
12: **end function**

---

**Latent Step.** During the first half of the sampling process, we update the latent variables. We establish a loss threshold of 0.2; once this threshold is reached, no further updates are made.

## I BASELINES

**Token Re-weighting.** Token Re-weighting method manually adjusts the weights of certain tokens to control their influence.

Specifically, a CLIP text encoder embeds the input text into a sequence of tokens $s = \{v_1, \ldots, v_K\}$. Token Re-weighting multiplies a scalar $\alpha$ with specific embeddings, for example, modifying the first token to $s' = \{\alpha v_1, \ldots, v_K\}$. The updated sequence is then used as a new conditioning input for the diffusion model. This is implemented by the `compel` package.

## J LIMITATIONS

Although Vico effectively allocates attribution across different tokens, it does not explicitly bind attributes to subjects. Moreover, there is a critical balance to maintain between latent updates and semantic coherence. Excessive updating can lead to the generation of nonsensical videos.

## K    BROADER APPLICATIONS

Technically, the computation of attention flow proposed in our system is versatile and can be efficiently applied to a variety of other applications like erase certain concept in diffusion models. Additionally, the principle of fairly distributing the contribution of different input parts can be extended to other domains, such as language modeling.

