# OpenReview forum: "Compositional Video Generation as Flow Equalization"
_ICLR.cc/2025/Conference — Submitted to ICLR 2025_

### Official Review · Reviewer_kAsb · 2024-11-01

**Soundness:** 3
**Presentation:** 2
**Contribution:** 2
**Rating:** 5
**Confidence:** 4

**Summary:**

This paper presented a new method named Vico, a framework designed for compositional video generation. Different from existing methods that may not reflect the intended composition of elements, the proposed Vico tries to adjust the model to ensure that no single concept dominates. Specifically, Vico calculate the contribution of each text token using max-flow and leverage a sub-graph flow to propagate information. The proposed method can be implemented by inserting into diffusion-based methods. Extensive experimental results verify the effectiveness of the proposed method.

**Strengths:**

1.	A new method named Vico for compositional video generation is proposed.
2.	The proposed Vico leverage max-flow to ensure that no single concept dominates.
3.	Extensive results verify the effectiveness of the proposed method in generation high-quality videos.

**Weaknesses:**

1.	This paper argues that the proposed method explicitly constrains the contribution of each token to be the equal, but in fact there are some words with no contribution or words with outstanding contribution in a sentence, that is, the importance of each word should be different. For example, "On a certain day, a boy is running", where "On a certain day" is not informative, only the following "the boy is running" is informative. The authors should discuss the problem.
2.	The proposed method needs to be optimized at test time, and the resource consumption of optimization such as time and computational overhead should be discussed.
3.	This paper said “While the compositional text-to-image sythesis Liu et al. (2022); Chefer et al. (2023); Kumari et al. (2023); Feng et al. (2023); Huang et al. (2023) has been more studied, the challenge of compositional video generation has received less attention.” My question is that whether these image methods can be extended to compositional video generation task? If not, the authors should give a explanation. If yes,  please give a comparison with them.
4.	Can Vico be used in compositional motion scenarios? Such as "In a room, a cat is running from left to right, a dog is running from right to left", this complex movement scene is very interesting.

--------------------------------
Post review:
Thank the authors for the responses. Given ICLR's high standards, I decided to keep the score at 5. The motivation of this paper is good, but in the current version, the different processing of prompt words may ignore some cases, such as when some words other than verbs and nouns have a certain amount of information. And the proposed method requires test-time optimization, which greatly increases the inference cost, considering that the model is getting larger and larger now.

**Questions:**

Please refer to Weaknesses for more details.

---

> ### Author Response · Authors · 2024-11-20
> **Thank Reviewer kAsb for the suggestions**
>
> We greatly appreciate Reviewer kAsb's thoughtful suggestions and have thoroughly incorporated them into the revised manuscript.
>
> `>>> Q1` **Not all token are equal**
>
> `>>> A1` We are sorry for any confusion this may have caused. The reviewer is right; some tokens indeed carry less information.
>
> Vico does not assume that all tokens require balancing. Instead, our approach focuses on assigning equal weight to **specific important tokens**. As mentioned in `Line 417`, we only equalize `nouns and verbs`.
>
> Based on your suggestion, we have revised the manuscript `Section 3.1` to emphasize this point more clearly.
>
> `>>> Q2` **Resource Consumption**
>
> `>>> A2` We truly thank the reviewer for this question. We have incorpate the running speed and complexisty in `Appendix G`. Regarding memory consumption, for generating a 16-frame video at $320\times 512$ resolution with VideoCrafterv2, the results are as follows:
>
> |Method| GPU Mem.|
> |--|--|
> |VideoCrafterv2| 12G |
> |VideoCrafterv2+Vico | 29G |
>
> The GPU memory usage increases by $\times 2.4$. We are actively working on optimizing and reducing this.
>
> `>>> Q3` **Extend Image Methods to Vidoe**
>
> `>>> A3`  Excellent question. Yes, image-based methods can be extended to compositional video generation tasks. In fact, we have implemented *Token Re-weight* and *Attend-and-Excite* and compared their performance with Vico, as shown in `Figure 4` and `Table 1` of the paper. However, their results do not match Vico's performance.
>
> The main reason for the gap is the difference in model architectures and the need to handle temporal dynamics. This is explained in `Lines 200-215`, where we show why we need new design for video tasks.
>
> `>>> Q4` **Compositional Motion Scenarios**
>
> `>>> A4` Yes, Vico can handle compositional motion scenarios.
>
> **Quantitative Results**: We introduced the *Motion Composition* metric in `Line 371`. Column 4 of `Table 1` shows our results. Vico improves motion composition by handling different movements for multiple subjects.
>
> **New Cases**: We have added new examples of this scenario in `Appendix E.1`. These samples highlight Vico's improved performance in complex motion cases.

---

> > ### Comment · Reviewer_kAsb · 2024-11-25
> >
> > Thank the authors for the responses.  However, the test time of the proposed method is still missing. And I am still confused about the equalization of nouns and verbs, as many other words may also be useful.

---

> > > ### Author Response · Authors · 2024-11-29
> > >
> > > Dear Reviewer kAsb,
> > >
> > > Thank you for your thoughtful suggestions and comments. We have added the test time details and tried to clarify the equalization of nouns and verbs as you suggested.
> > >
> > > Please let us know if there’s anything else we should address or if you have any further questions.
> > >
> > > Thank you again for your time!

---

> ### Author Response · Authors · 2024-11-25
>
> We sincerely thank Reviewer kAsb for the insightful questions.
>
> `>>> Q1 ` **Test time of Vico**
>
> `>>> A1` Originally presented in `Appendix G, Table 9`, we report test time based on a 50-step DPM denoising process using the VideoCrafterv2 model at a resolution of 512 × 320 for 16 frames, running on a single A6000 GPU.
>
> The time is also reported below. Specifically, the baseline VideoCrafterv2 completed the task in 23 seconds, while our Vico took 45 seconds. Despite its additional complexity, Vico’s efficient design keeps the inference time within a reasonable range.
>
> | Method              | Time  |
> |---------------------|-------|
> | VideoCrafterv2      | 23s   |
> | + Attend&Excite     | 48s   |
> | + Vico (soft&hard)  | 45s   |
>
> `>>> Q2 ` **Selection of key words**
>
> `>>> A2` Thank the reviewer for raising this concern. We totally understand the doubts about selecting only `nouns and verbs`, which may seem overly simplistic. However, this approach aligns with our core evaluation metrics in the paper.
>
> Our primary metrics—**Spatial Relations, Multiple Objects, and Motion Composition**—ultimately assess the accuracy of `nouns and verbs` in the generated videos. For this reason, we rely on this method at this stage.
>
> **Extensions and Improvements**
>
> While this approach works well for now, we acknowledge its limitations and are open to enhancements.  For instance, advanced keyword extraction techniques like prompting LLM could be employed to enhance keyword selection. We provide an [example](https://chatgpt.com/share/6744030a-db00-8002-923e-85c6211d0864) on how to use this.
>
> > User: Extract the key action or concept from the sentence for use in a text-to-video generative model. Provide only the answer. Example sentence: 'On a certain day, a boy is running.'
>
> > ChatGPT: a boy is running.
>
> Such techniques can be incorporated into our framework to enhance its robustness and flexibility when selecting keywords.
>
> ----------
>
> Should you have any further questions or concerns, please let us know and we will try our best to clarify.
>
> Again, thank your for all the suggestions and questions—they've really helped us to improve our paper!

---

> > ### Author Response · Authors · 2024-11-26
> >
> > Dear Reviewer kAsb,
> >
> > We would like to thank you again for your constructive comments and kind effort in reviewing our submission.
> >
> > Please let us know if our response addressed your concerns. We are happy to address any further feedback.
> >
> > Thanks!

---

### Official Review · Reviewer_Y7gd · 2024-11-01

**Soundness:** 3
**Presentation:** 2
**Contribution:** 2
**Rating:** 6
**Confidence:** 5

**Summary:**

This paper introduces Vico, a novel framework specifically designed for compositional video generation. Vico employs a maximum flow approach to ensure fair contributions from each input token, integrating sub-graphs, a soft flow strategy, and vectorized path flow computation for efficient inference. Extensive experiments demonstrate that Vico significantly surpasses existing baselines in compositional generation tasks.

**Strengths:**

1. This paper presents a highly innovative solution to the problem, utilizing traditional max flow to address token-level response balancing, thereby achieving effective compositional generation.
2. I appreciate that extensive effort has been put into designing feasible experiments. The authors introduce practical techniques such as subgraph, soft min, and vectorized flow strategies, which significantly enhance inference speed.
3. The experiments are thorough and well-executed, including comprehensive comparative studies and a detailed user study, which provides strong validation for the proposed approach.
4. The visual quality of the generated videos is satisfying, despite the limited number of examples presented.

**Weaknesses:**

1. The primary concern is the lack of comparisons or discussions involving recent text-to-video (T2V) methods. The baseline model, VideoCrafter2, was released over a year ago. To convincingly demonstrate the relevance of the compositional generation problem, the paper should ideally compare against more advanced, recent baselines like OpenSora[1],  CogVideoX[2], or more.
2. Additionally, the paper lacks comparisons to existing compositional video generation models. For instance, methods like LVD[3], which uses LLMs for prompt decomposition with gradient-based layout optimization during inference, VideoTetris[4], which employs spatiotemporal diffusion for multi-object generation, and VideoDirectorGPT[5], which combines an LLM director with spatial convolutions for layout learning, are not discussed. Including comparative studies with these related works would strengthen the evaluation and clarify the advantages of the proposed approach.
3. The paper lacks a theoretical justification for the assumption that each token should have an equal impact on the generated result. In typical prompts, not all tokens necessarily require equal influence; for instance, non-descriptive tokens or function words might logically play a reduced role in the attention process. Given the strength of this assumption, a more rigorous rationale or proof would help validate its applicability.
4. Another strong assumption is made regarding cross-attention in T2V models. The paper’s approach relies on cross-attention being embedded uniformly across all layers and frames, yet many contemporary T2V models, such as CogVideoX[2], Mochi-1[6], and commercial models like Kling[7], utilize 3D attention mechanisms similar to MMDiT. This raises concerns about the applicability of the paper’s assumptions to modern architectures, which may limit its generalizability.
5. The paper contains a few typographical errors that affect readability. For example, in the introduction, the prompt "a bird looks like a cat" is used as an example, but the actual effect of this prompt is not demonstrated.

[1] Open-Sora: Democratizing Efficient Video Production for All, https://github.com/hpcaitech/Open-Sora

[2] Yang, Zhuoyi, et al. "Cogvideox: Text-to-video diffusion models with an expert transformer." arXiv preprint arXiv:2408.06072 (2024).

[3] Lian, Long, et al. "Llm-grounded video diffusion models." arXiv preprint arXiv:2309.17444 (2023).

[4] Tian, Ye, et al. "VideoTetris: Towards Compositional Text-to-Video Generation." arXiv preprint arXiv:2406.04277 (2024).

[5] Lin, Han, et al. "Videodirectorgpt: Consistent multi-scene video generation via llm-guided planning." arXiv preprint arXiv:2309.15091 (2023).

[6] Mochi-1: https://www.genmo.ai/blog

[7] Kling: https://www.klingai.com/

**Questions:**

1. My primary question is,  do the paper's assumptions hold true for modern 3D attention models, such as the open-source CogVideoX. Can the proposed method balance token contributions effectively when applied to an attention graph in such models?
2. In the case of 3D attention, how much additional computational overhead does the proposed approach introduce? Cross-attention calculations are relatively lightweight, but if full attention is employed, as in MMDiT, would the method’s computations lead to significant cost increases?
3. The paper mainly showcases relatively short prompts. I would like to ask if the proposed method can handle modern, highly descriptive prompts that may reach up to 100 words. Would the method remain feasible for inference, and what would be the associated computational cost in such cases?
4. As a fair comparison, could the paper utilize T2V CompBench[1], a benchmark specifically designed to evaluate compositional video generation, to better assess and compare the performance of the proposed model against other existing models? I wonder how well this method performs under a more comprehensive benchmark.

[1] Sun, Kaiyue, et al. "T2V-CompBench: A Comprehensive Benchmark for Compositional Text-to-video Generation." arXiv preprint arXiv:2407.14505 (2024).

---

> ### Author Response · Authors · 2024-11-20
> **Thank Reviewer Y7gd and Response (Part 1)**
>
> We truly thank the review for all the questions. They are very important to improve the practical application of our Vico to more video diffusion model.
>
> **Note**: Kindly reminder that, according to this year's [ICLR policy](https://iclr.cc/Conferences/2025/FAQ), papers published (peer-reviewed) after **July 1, 2024**, are treated as concurrent work and `authors are not required to compare their own work to that paper`. References [5] and [4] were accepted to COLM24 and NeurIPS24, respectively, with announcements made after July 1. Additionally, Cogvideox [2] was released in August and T2V-CompBench [1] was released in July, and [6] and [7] are not research papers. We kindly ask the reviewer to consider these circumstances.
>
> `>>> Q1` **Recent Baseline Models**
>
> `>>> A1` We truly thank the review for the suggestions. As suggested, we built Vico on top of Open-sora [1] and CogVideoX [2], and discussed in `Appendix F`.
>
> - **Open-sora Adaptation**: Open-sora [1] was relatively straightforward to adapt, as it uses STDiT, which separates spatial and temporal attention.
>
> - **CogVideoX Adaptation**: Interestingly, our solution also applies well to CogVideoX. It uses a more complex 3D MM-DiT architecture. This model flattens all text and video tokens, processing them in a single joint attention layer. Since there is no explicit cross-attention mechanism, traditional cross-attention control methods don’t apply. However, our graphical abstraction approach is highly effective in this scenario because, fundamentally, it is still attention between tokens. Specifically, we redefine our graph construction rule as follows:
>
>    $$\mathbf{W}\_{l} = \begin{bmatrix} E_{tt,l}  & E_{tv,l} \\\\
>     E_{vt,l} & E_{vv,l} \end{bmatrix}$$
>    $$\mathbf{W} =
>    \begin{bmatrix} \mathbf{W}_1  & \mathbf{0} & \dots &  \mathbf{0} \\\\
>    \mathbf{0} &\mathbf{W}_2 & \dots & \mathbf{0} \\\\
>    \mathbf{0} &\mathbf{0} & \vdots & \mathbf{0} \\\\
>    \mathbf{0} &\mathbf{0} & \vdots & \mathbf{W}_L \end{bmatrix}$$
>
>    Here, $\mathbf{W}\_l$ is the adjacency matrix at each layer, with $E_{tt,l}$, $E_{tv,l}$, $E_{vt,l}$, and $E_{vv,l}$ representing weights for text-to-text, text-to-video, video-to-text, and video-to-video connections, respectively. Each $E$ is still subject to our computation defined from `Line 236`. Stacking these matrices across layers gives the final capacity matrix $\mathbf{W}$.
>
> **Results**: We tested Vico with new baselines on VBench, measuring the *Multiple Object Composition* score. Due to the high memory demands for DiT, we used an 80GB A100 GPU for inference.
>
> The results show that Vico substantially enhances performance, proving the effectiveness of our approach across various architectures.
>
> |Method|Multiple Object|
> |--|--|
> |Open-Sora|33.64|
> |+Vico|**48.21**|
> |CogVideoX 2B|53.70|
> |+Vico|**63.21**|
>
> We have included the results in the revise paper `Appendix F`.
>
> `>>> Q2` **Recent work on compositional video generation**
>
> `>>> A2` We thank the R-Y7gd for bring those nice reference. We have add them to our reference and discuss with their difference. We differ from them in two main perspectives,
>
> - **Training-free**. Methods like [3][5] require curated datasets with bounding boxes and prompts, while [4] relies on long-video datasets with enhanced motion dynamics. In contrast, Vico is fully training-free, using only test-time optimization without additional training data.
> - **Additional components**: [3][5] depend on external LLMs for spatial planning, and [4] introduces a Reference Frame Attention module and optionally uses LLMs for prompt decomposition. Vico requires no additional modules or external support.
>
> **Comparing with new papers**:
> As suggested, we compare our VC2+Vico model with those new baselines on `T2V-CompBench`. The results is shown below. Even without reliance on external datas, models and training, we can gets comparable performance, and even better on **action binding** and **object interactions**.
>
> We have also included the new results in the main paper `Table 2`.
>
> |Model| Consist-attr| Spatial | Motion |Action | Interaction |
> |--|--|--|--|--|--|
> | LVD | 0.5595 | **0.5469** | **0.2699** | 0.4960 | 0.6100 |
> | VideoTetris | **0.7125** | 0.5148 | 0.2204 | 0.5280 | `0.7600` |
> |VC2+Vico | `0.6980` | `0.5432` | `0.2412`| **0.6020** |**0.7800** |
>
> `>>> Q3` **Equal influence justification**
>
> `>>> A3` Thanks for the nice question. The reviewer is correct that not all tokens should be equally weighted.
>
> Our approach does not assume that all tokens need balancing; rather, we focus on giving equal weight to specific important tokens. As mentioned in `Line 417`, we only equalize `nouns and verbs for equalization`.
>
> We have also revise `Section 3.1` to highlight this point.

---

> ### Author Response · Authors · 2024-11-20
> **Thank Reviewer Y7gd and Response (Part 2)**
>
> `>>> Q4` **Vico on 3D attention**
>
> `>>> A4` This is an excellent question. Vico does not strictly require cross-attention. In fact, it can be seamlessly applied to recent diffusion transformer models with 3D attention. The only adjustment needed is in constructing the capacity matrix.
>
> As implemented in `A1`, we have successfully apply Vico to 3D attention.
>
>
> `>>> Q5` **'Bird/Cat' Example**
>
> `>>> A5` Sorry for the confusion. In fact, the example can be found in `Figure 1` with prompt `A bird and a cat`.
>
> In `Line 51`, we use the phrase `a bird looks like a cat` to explain how Videocrafterv2 mistakenly places a cat’s face on the bird’s body. We did not mean to imply there was an actual prompt `a bird looks like a cat`.
>
> We have revised the text for better clarity.
>
>
> `>>> Q6` **Vico on 3D attention**
>
>
> `>>> A6` Yes. Please see `A1` and `A4`.
>
> `>>> Q7` **Computation when applying Vico to 3D attention**
>
> `>>> A7` The computation is large, still manageable.
>
> - **Forward Consumption**: Assume each MM-DiT layer has $n$ tokens (including all text and video tokens), in the end of the day, computing our approximated ST-Flow for $L$ layers is $Ln^3$.
>
>    For CogVideoX-2B, generating a 512x512 video with 49 frames involves 13,312 video tokens and fewer than 226 text tokens, totaling roughly 13,000 tokens. Using FP16 precision, this adds approximately 10GB of GPU memory to the original model's inference.
> - **Backward Consumption**: Since we only need to back-propagate to the input, we set all network parameters to `require_grad=False`.
>
> Altogether, we need ~76GB of GPU memory for CogVideoX-2B inference using FP16, which fits within a single A100 GPU. The running time is ~200s for each prompt.
>
> `>>> Q8` **Vico on Long Prompt**
>
>
> `>>> A8` The current token length is primarily limited by the CLIP text encoder used in the diffusion model, which typically defaults to 77 tokens.
>
> As suggested, we test on longer prompts, and show the results in `Appendix E.2`. The findings show that Vico can still generate coherent videos even with longer input texts.
>
> `>>> Q9` **Results on T2V CompBench**
>
> `>>> A9` We appreciate the suggestion for new brilliant benchmark and we have add the reference in the paper.
>
> As suggested, we evaluated our method on this new benchmark and compared it with several recent baselines. As shown below, combining VideoCrafterv2 with Vico delivers competitive performance on this benchmark.
>
> |Model| Consist-attr| Spatial | Motion |Action | Interaction |
> |--|--|--|--|--|--|
> | LVD | 0.5595 | **0.5469** | **0.2699** | 0.4960 | 0.6100 |
> | VideoTetris | **0.7125** | 0.5148 | 0.2204 | 0.5280 | `0.7600` |
> |Open-Sora 1.2| 0.6600| 0.5406| 0.2388 |0.5717 |0.7400 |
> |VideoCrafterv2 | 0.6750 | 0.4891 | 0.2233| `0.5800` |`0.7600` |
> |VideoCrafterv2+Vico | `0.6980` | `0.5432` | `0.2412`| **0.6020** |**0.7800** |

---

> > ### Comment · Reviewer_Y7gd · 2024-11-24
> > **Thanks for your response.**
> >
> > Thanks for the author's detailed explanation and the additional experiments. I believe the paper is now much more complete, and I have raised my score to 5.
> > I still have a few questions that I find interesting, and I hope addressing them could further enhance the paper’s impact:
> >
> > **Q1: Qualitative Experiments on CogVideoX**
> >
> > Could you provide some qualitative experiments on CogVideoX? Showing specific cases would be very helpful.
> > As models like CogVideoX have become mainstream, demonstrating a few concrete cases could effectively showcase Vico’s practical performance and highlight its advantages more clearly. I believe in Appendix F, only quantitative results are reported.
> >
> > **Q2: How to integrate Vico framework in training?**
> >
> > Currently, the inference time seems to be quite high. For instance, while CogVideoX-2B takes 90s to generate a sample on A100 80G, Vico requires 200s, effectively doubling the inference cost.
> >
> > Is there a way to move some parts of the attention flow process into the training phase to reduce deployment time?
> > Optimizing the inference time could significantly enhance Vico's applicability and practical potential.

---

> > > ### Author Response · Authors · 2024-12-01
> > >
> > > Dear Reviewer Y7gd,
> > >
> > > We would like to thank you again for your constructive comments and kind effort in reviewing our submission.
> > >
> > > Please let us know if our new response addressed your concerns. We are happy to address any further feedback.
> > >
> > > Thanks!

---

> > > > ### Comment · Reviewer_Y7gd · 2024-12-03
> > > > **Thanks for you response**
> > > >
> > > > I appreciate the new results provided. My concerns have been thoroughly addressed, and I have updated my score to a 6. I believe incorporating these additional experiments and comparisons into the camera-ready version would make the paper more comprehensive and impactful. I look forward to the practical application of Vico in training.

---

> ### Author Response · Authors · 2024-11-24
>
> We truly thank the reviewer for the thoughtful questions and feedback.
>
> `>>> Q1` **Qualitative Experiments on CogVideoX**
>
> `>>> A1` Thank R-Y7gd for the suggestion. As requested, we selected representative samples generated by CogVideoX-2B and CogVideoX-2B+Vico from the *Multiple Object* subset of Vbench. The results are included in the revised  `Appendix F, Figure 7`.
>
> Our observations indicate that **compositional errors persist even in advanced video diffusion models** like CogVideoX. For instance, it blends *a boat and an airplane* into a single object, such as a *seaplane*, or generates only *a pizza* while neglecting *a tie*. In comparison, Vico resolves these conflicts and ensure all concepts are represented accurately and fairly.
>
> `>>> Q2` **How to integrate Vico framework in training?**
>
> `>>> A2` We sincerely thank the reviewer for this excellent question; this is something we are actively working on.
>
> **Compositional Regularization**: One approach we are considering is incorporating the *Vico objective function as a regularization loss during training*. Given a noisy latent $\mathbf{x}_t$ at step $t$ and the text $y$, we can compute its attribution score $A$ during a training iteration. The diffusion model can then be optimized simultaneously for denoising and balancing token influence.
>
> **Synthetic Videos**: Another potential approach is to *use Vico to generate videos containing compositional concepts*. These synthetic samples can then be used to retrain the diffusion model, improving its ability to handle complex compositions.
>
> In general, test-time optimization is indeed lengthy and costly. Moving it to the training phase is indeed a promising direction to improve efficiency.
>
> ----------
>
> Once again, thank the reviewer for the thoughtful feedback and for recognizing the potential of our work.

---

### Official Review · Reviewer_JUy3 · 2024-11-04

**Soundness:** 3
**Presentation:** 2
**Contribution:** 3
**Rating:** 8
**Confidence:** 4

**Summary:**

This manuscript tackles the problem of video generation, focusing on improving the compositional and complex interactions in the final output. The proposed approach utilizes a flow equalization formulation to ensure that different tokens of input get a fair amount of attention throughout the self-attention layers.

**Strengths:**

Capturing compositional relationships in the final generated output is a very important problem and, to the best of my knowledge, is one of the biggest issues with current SOA in GenAI.
This paper correctly identifies one of the main issues throughout the attention mechanism and tries to improve the contribution of different tokens in attention layers as a test-time optimization. The authors model the information flow of the generative model as a graph, which is a smart and (semi-)novel strategy (for the videos) in my opinion.

**Weaknesses:**

The biggest weakness of the solution is the readability of this paper. It was very hard for me to read through the text and jump from text to mathematical notations and back. I will ask for a few clarifications in the questions block.

**Questions:**

1-  In line 247, the capacity matrix W, what are each row and column? Why does the first row have the `t` index while other rows don't?
2- Lines 241, and 243. What is the difference between e_(i,j), i=j and e_(i,i) difference?
3- Section 3.1: At first glance, it is not clear what is x_t (definition) and how A_i is extracted out of it.
4- What is token-reweight in Table 1?
5- There are some valid categories of issues in Video Generation shown in Figure 1. My question is if the lack of fair attribution and attention is the only problem that causes these issues. If not, what are the other factors?

---

> ### Author Response · Authors · 2024-11-20
>
> We sincerely appreciate Reviewer JUy3's suggestions. We have carefully incorporated them into the revised manuscript.
>
> `>>> Q1` **Readability**
>
> `>>> A1` Thank R-JUy3 for the suggestion. We sincerely apologize for any issues with the writing. The paper adapts many mathematical definitions to a new problem, which may have caused confusion. We have revised the paper to address this and improve clarity.
>
>
> `>>> Q2` **Capacity matrix W**
>
> `>>> A2` Thanks for the question.
> - **Row and Column**: As shown in `Line 250`, our $\mathbf{W}$ is written as a form of **block matrix**. Each $E_i$ and $E_{t,i}$ itself is a smaller matrix, with each element in them representing the weight $e$ in the edge.
> - **$t$ in first row**: Only the first row has `t` because that, the **text tokens are positioned only in the first row**. The rest of rows are video tokens. In this sense, the matrix $E_{t,l}$ represents the influence from text tokens to video tokens at each cross-attention layer. $E_{l}$ stands for the self-attention connections at the $l$-th layer, which only connect between video tokens.
>
> We have included this explanation in the revised version to clarify the structure of $\mathbf{W}$ in `Line 253-257`.
>
> `>>> Q3` **$e_{i,j}$ when $i=j$ and $e_{i,i}$**
>
> `>>> A3` Sorry for the confusion. Yes, $e_{i,j}$ when $i=j$ and $e_{i,i}$ are the same; they represent the same value using different notations.
>
>
> `>>> Q4` **$\mathbf{x}_t$ and $A_i$**
>
> `>>> A3` Thank the reviewer for the question. $\mathbf{x}_t$ stands for the noisy latent at timestep $t$, which is defined in `Equation 1`.
>
> $A_i$ is the importance for each token. We make it a general definition because there could be other measurement of $A_i$. In this paper, we define it as the flow value from each token to the final video, which is in `Sec 3.2` and `Sec 3.3`. A simple intuition is that, $A_i$ is the accumulated weight of all cross-attention and self-attention score through all layers, from text token $v_i$.
>
> We have added new explaination in the `Sec 3.1`.
>
> `>>> Q4` **Token Re-weighting**
>
> `>>> A3` Thanks for the nice question. Token Re-weighting method manually adjusts the weights of certain tokens to control their influence.
>
> Specifically, a CLIP text encoder embeds the input text into a sequence of tokens $s = \{v_1, \dots, v_K\}$. Token Re-weighting multiplies a scalar $\alpha$ with specific tokens, for example, modifying the first token to $s' = \{\alpha v_1, \dots, v_K\}$. The updated sequence is then used as a new conditioning input for the diffusion model. As mentioned in `Line 354`, it is implemented by the `compel` package.
>
> As suggested, we add the explaination in the revise `Appendix I`.
>
>
> `>>> Q5` **Factors for compositional issues**
>
> `>>> A5` Great question. Actually, fair attribution is not the only reason for compositional failures.
>
> Specifically, fair attribution is key for `first-order` composition, which means that all concepts are represented in the generated output.
>
> However, for `second-order` or `high-order` composition, other types of errors may still occur. For instance, even when all concepts are fairly presented, attributes might be incorrectly assigned or mis-bidded [A] with objects. A common example would be generating "*a red dog and a yellow cat*" from a prompt of "*a red cat and a yellow dog*".
>
> Our primary focus is on solving simpler cases first. We are working to extend our approach to handle more complex problems.
>
> > [A] Linguistic Binding in Diffusion Models: Enhancing Attribute Correspondence through Attention Map Alignment, NeurIPS 2023

---

> > ### Comment · Reviewer_JUy3 · 2024-11-26
> > **Thanks for the answers**
> >
> > Thanks for the clarifications. I keep my score unchanged.

---

> > > ### Author Response · Authors · 2024-11-26
> > >
> > > Thank Reviewer JUy3 so much for your kind words and for taking the time to review our paper.
> > >
> > > Your feedback and support mean a lot to us!

---

### Official Review · Reviewer_6LSE · 2024-11-04

**Soundness:** 3
**Presentation:** 3
**Contribution:** 3
**Rating:** 8
**Confidence:** 3

**Summary:**

This paper introduces a novel approach for compositional video generation. The proposed method begins by analyzing the impact of input tokens on the video output, ensuring that no single concept predominates the generated content. The core concept involves computing each text token's contribution to the video generation process through maximum flow. The paper introduces an efficient computational technique by approximating the subgraph flow with a vectorized implementation.Finally, the method has been tested across various diffusion-based video models, and the experimental results confirm its effectiveness in enhancing both visual fidelity and semantic accuracy.

**Strengths:**

1. The proposed method is innovative and can be integrated with existing video generation techniques. The experiments demonstrate the effectiveness of applying the "Vico" method on current models such as AnimaDiff, ZeroScore V2, and VideoCrafter V2, showing notable improvements in results.

2. The paper is well-articulated, featuring thorough theoretical analysis and proof.

**Weaknesses:**

1. The user study (Table 2) is limited to only ten video clips, which is insufficient to conclusively prove the effectiveness of the method.

**Questions:**

n/a

---

> ### Author Response · Authors · 2024-11-20
> **Thank Reviewer 6LSE for the questions**
>
> We truly thank the R-6LSE for the nice comments.
>
> `>>> Q1` **User Study**
>
> `>>> A1` Great suggestion. As suggested, we conducted a new user study using **50 video clips** generated from VideoCrafterv2 with prompts from VBench. We generated attribution heatmaps using Cross-Attention, Attention Rollout, and our ST-Flow. Additionally, we increased the **number of participants to 43**, who rated each method from 1 to 5 on *Temporal Consistency* and *Reasonability*.
>
> The results is shown below as the mean$\pm$std. It illustrates that, our ST-Flow consistently outperformed the other methods.
>
> We have included this results in the manuscript `Table 3`.
>
> |Attribution Method | Temporal Consistency&uarr;| Reasonability&uarr;|
> |----|---|---|
> |Cross-Attention | 2.62$\pm0.12$ | 2.87$\pm0.23$ |
> |Attention Rollout | 3.77$\pm0.20$ | 3.36$\pm0.28$ |
> |ST-Flow (Ours) | 4.12$\pm0.13$ | 3.76$\pm0.19$ |

---

### Author Response · Authors · 2024-11-20
**Thank all reviewers for their constructive feedback**

We sincerely thank all reviewers for their constructive feedback. We deeply appreciate the following positive comments:

- The paper addresses an important problem: `Reviewer JUy3`
- The method is innovative, leveraging max-flow for token balancing: `Reviewer JUy3, Reviewer Y7gd, Reviewer 6LSE`.
- The experiments are thorough and well-executed: `Reviewer 6LSE, Reviewer Y7gd, Reviewer kAsb`.
- The video quality is impressive and compositionally accurate: `Reviewer Y7gd, Reviewer kAsb`.
- The writing is clear and supported by strong theoretical analysis: `Reviewer 6LSE`.

We will address the specific questions and concerns raised by the reviewers in the subsequent sections of this rebuttal.

---

### Meta-Review · Area_Chair_YjVS · 2024-12-19

**Metareview:**

This paper introduces Vico, an inference-time method designed to enhance compositional video generation. Vico ensures equal influence for each textual token (verbs and nouns only) on the final video output through test-time optimization, dynamically assessing and rebalancing token impacts at each diffusion step. The reviewers recognize the significance of the problem, the (semi-)novel approach, and the extensive results that demonstrate the method's effectiveness.

However, the reviewers also highlight several weaknesses. While many concerns were addressed following the rebuttal, the remaining issues are:

* Test-time optimization overhead: Two reviewers raised concerns about the significant increase in computation time. For instance, Vico requires 200 seconds for CogVideoX-2B, more than doubling the 90 seconds needed by the baseline CogVideoX. The authors responded by outlining plans to integrate the Vico framework into the training phase to mitigate this overhead.

* Balancing strategy: Reviewer kAsb questioned the approach of equalizing only nouns and verbs, suggesting it may oversimplify the process.

Finally, this paper receives mixed evaluations: accept (low confidence), marginally above the acceptance threshold, and marginally below the acceptance threshold. The opposing Reviewer kAsb remains unwilling to adjust their score after reading the rebuttal, making it a borderline case.

The AC finds Reviewer kAsb's arguments persuasive and agrees with their concerns. While this work shows potential for improvement, the significant overhead introduced by test-time optimization diminishes its practical significance. Regarding the technical design, the AC agrees with Reviewer kAsb's critique that balancing only nouns and verbs may overlook other important word types. This limitation could hinder the model's ability to effectively handle long and complex prompts. This limitation is particularly relevant as recent video generation models often use rephrasers to expand short user prompts into complex structures, incorporating multiple nouns, verbs, and other word types (e.g., attributes, geometric details, relationships) that convey subtle yet essential information.

In light of the above, the AC recommends that accepting this submission may be premature until the concerns are fully resolved.

**Additional Comments On Reviewer Discussion:**

Two reviewers responded to the rebuttal: one increased their score to 6, while the other remained unwilling to change their score of 5. The points raised by the reviewers, along with the AC's assessment, are detailed above in the metareview.

---

### Decision · Program_Chairs · 2025-01-22

Reject